# Ancestry and demography and descendants of Iron Age nomads of the Eurasian Steppe

Martina Unterländer[1,*], Friso Palstra[2,*], Iosif Lazaridis[3,4], Aleksandr Pilipenko[5,6,7], Zuzana Hofmanová[1], Melanie Groß[1], Christian Sell[1], Jens Blöcher[1], Karola Kirsanow[1], Nadin Rohland[3], Benjamin Rieger[8], Elke Kaiser[9], Wolfram Schier[9], Dimitri Pozdniakov[5], Aleksandr Khokhlov[10], Myriam Georges[2], Sandra Wilde[1], Adam Powell[1,11], Evelyne Heyer[2], Mathias Currat[12], David Reich[3,4,13], Zainolla Samashev[14], Hermann Parzinger[15], Vyacheslav I. Molodin[5,6] & Joachim Burger[1]

During the 1st millennium before the Common Era (BCE), nomadic tribes associated with the Iron Age Scythian culture spread over the Eurasian Steppe, covering a territory of more than 3,500 km in breadth. To understand the demographic processes behind the spread of the Scythian culture, we analysed genomic data from eight individuals and a mitochondrial dataset of 96 individuals originating in eastern and western parts of the Eurasian Steppe. Genomic inference reveals that Scythians in the east and the west of the steppe zone can best be described as a mixture of Yamnaya-related ancestry and an East Asian component. Demographic modelling suggests independent origins for eastern and western groups with ongoing gene-flow between them, plausibly explaining the striking uniformity of their material culture. We also find evidence that significant gene-flow from east to west Eurasia must have occurred early during the Iron Age.

[1] Palaeogenetics Group, Institute of Evolutionary Biology, Johannes Gutenberg University Mainz, 55099 Mainz, Germany. [2] CNRS UMR 7206 Eco-anthropologie, Muséum National d'Histoire Naturelle, 75016 Paris, France. [3] Department of Genetics, Harvard Medical School, Boston, Massachusetts 02115, USA. [4] Broad Institute of MIT and Harvard, Cambridge, Massachusetts 02142, USA. [5] Institute of Cytology and Genetics, Siberian Branch, Russian Academy of Science, Akademika Lavrentieva 10, Novosibirsk 630090, Russia. [6] Institute of Archaeology and Ethnography, Siberian Branch, Russian Academy of Science, Akademika Lavrentieva 17, Novosibirsk 630090, Russia. [7] Novosibirsk State University, Pirogova str. 2, Novosibirsk 630090, Russia. [8] Molecular Genetics and Genome Analysis Group, Institute of Evolutionary Biology, Johannes Gutenberg University Mainz, 55099 Mainz, Germany. [9] Department of History and Cultural Studies, Freie Universität Berlin, 14195 Berlin, Germany. [10] Samara State University of Social Sciences and Education, Samara 443099, Russian Federation. [11] Max Planck Institute for the Science of Human History, Kahlaische Straße 10, 07745 Jena, Germany. [12] Dépt. de Génétique & Evolution, Unité d'anthropologie, Université de Genève, 1205 Genève, Suisse. [13] Howard Hughes Medical Institute, Harvard Medical School, Boston, Massachusetts 02115, USA. [14] Branch of Margulan Institute of Archaeology, Astana 010000, Kazakhstan. [15] Stiftung Preussischer Kulturbesitz, 10785 Berlin, Germany. * These authors contributed equally to this work. Correspondence and requests for materials should be addressed to J.B. (email: jburger@uni-mainz.de).

During the first millennium BCE, nomadic people spread over the Eurasian Steppe from the Altai Mountains over the northern Black Sea area as far as the Carpathian Basin[1]. The classical Scythians, who had lived in the North Pontic region since the seventh century BCE, are the most famous among them due to the early reports in the *Histories* of Herodotus (490/480–424 BCE)[2]. Greek and Persian historians of the 1st millennium BCE chronicle the existence of the Massagetae and Sauromatians, and later, the Sarmatians and Sacae: cultures possessing artefacts similar to those found in classical Scythian monuments, such as weapons, horse harnesses and a distinctive 'Animal Style' artistic tradition. Accordingly, these groups are often assigned to the Scythian culture and referred to as 'Scythians'. For simplification we will use 'Scythian' in the following text for all groups of Iron Age steppe nomads commonly associated with the Scythian culture.

The origin of the widespread Scythian culture has long been debated in Eurasian archaeology. The northern Black Sea steppe was originally considered the homeland and centre of the Scythians[3] until Terenozhkin formulated the hypothesis of a Central Asian origin[4]. On the other hand, evidence supporting an east Eurasian origin includes the kurgan Arzhan 1 in Tuva[5], which is considered the earliest Scythian kurgan[5]. Dating of additional burial sites situated in east and west Eurasia confirmed eastern kurgans as older than their western counterparts[6,7]. Additionally, elements of the characteristic 'Animal Style' dated to the tenth century BCE[1,4] were found in the region of the Yenisei river and modern-day China, supporting the early presence of Scythian culture in the East. Artefacts of the Scythian culture spread over a large territory shortly after its emergence, but the underlying population dynamics that may have driven the cultural diffusion are poorly understood.

Genetic studies on Central Asian populations based on both ancient[8] and modern mitochondrial DNA (mtDNA)[9–12] agree that Central Asia has historically been a crossroad for population movements from east to west and vice versa. It has been claimed that gene flow occurred from east to west Eurasia as early as the Palaeolithic[13,14] and the Mesolithic[15], and from west to east Eurasia during the Bronze Age[16]. A recent genomic study[17] has emphasized the role of eastward migrations of people associated with the Yamnaya and Andronovo culture during the Bronze Age, followed by substantial admixture with East Asians. Most genetic studies on the later Iron Age nomads, however, have been limited by small sample size, restricted to a single cultural group, or based on the analysis of mtDNA alone[18–22].

In this study, mtDNA data from 96 individuals associated with the Scythian culture in different geographical regions and time periods have been sequenced and analysed; additionally, genomic data from eight of these individuals was obtained and analysed (Supplementary Table 1). From the western part of the Eurasian Steppe, samples discovered in the North Caucasus dating to the initial Scythian period (eighth to sixth century BCE), classical Scythians from the Don-Volga region (third century BCE), and Early Sarmatians from Pokrovka, southwest of the Ural (fifth to second century BCE), were included. From the eastern part of the Eurasian Steppe, we analysed samples from East Kazakhstan dating to the Zevakino-Chilikta phase (ninth to seventh century BCE); from the site Arzhan 2, assigned to the Aldy Bel culture in Tuva (seventh to sixth century BCE); and from the Tagar culture of the Minusinsk Basin (fifth century BCE). The majority of the samples generated by this study or retrieved from the literature date to the fourth to third century BCE and were discovered at archaeological sites situated in the Kazakh, Russian and Mongolian parts of the Altai Mountains. These findings were all assigned to the Pazyryk culture, which is named after the first discoveries by Gryaznov in 1927 and 1929 in the Pazyryk Valley

and famous for its rich frost-conserved graves, where human bodies, tapestry and clothing remained well-preserved[23–25] (Fig. 1).

While the eastern and western populations investigated here are separated by a distance of 2,000–3,500 km, archaeological evidence indicates that they were strikingly similar regarding their lifestyle and culture. The aims of this study are to investigate the extent to which these groups are genetically related to each other and whether they have a common origin, and to elucidate their demographic history and genetic relationships to modern living populations. We therefore divided the Iron Age steppe nomad data generated by this study and from the literature ($n = 147$, Supplementary Table 1) into seven sample groups (see Fig. 2) based on geographical and chronological criteria, and analysed these ancient individuals together with an extensive sample of modern individuals from 86 populations ($n = 3,410$, Supplementary Fig. 11 and Supplementary Table 19) from all parts of Eurasia. Our analysis included an array of statistical methods as well as a series of population genetic inference approaches, including explicit demographic modelling.

## Results

**Samples and sequence data.** We generated genome-wide capture data on a target set of 1,233,553 SNPs[26,27] for six individuals: two Early Sarmatians from the southern Ural region (PR9, PR3, group #3 in Fig. 2; fifth to second century BCE), two individuals from Berel' in East Kazakhstan (Be9, Be11, #6) dating to the Pazyryk period (fourth to third century BCE), and two individuals found in kurgan Arzhan 2 (A10, A17, #5) assigned to the Aldy Bel culture in Tuva (seventh to sixth century BCE). For Be9 and two additional individuals from east Kazakhstan (Is2 and Ze6, #4) dating to the Zevakino-Chilikta phase (ninth to seventh century BCE), we generated low coverage ($< 0.3$x) whole genome datasets by shotgun NGS (Table 1, Supplementary Tables 20 and 21).

We additionally obtained unambiguous and reproducible mtDNA sequences of the hypervariable region 1 (HVR1; bp 16,013–16,410) for 96 samples (out of 110 samples for which the analysis was attempted) (Supplementary Table 5). For 90/96 samples additional coding region SNPs were typed (Supplementary Table 6).

## Genetic relationship and origin of the Scythian groups.

The eastern sample group ($n = 113$) can be divided in four cultural subgroups chronologically dispersed over the 1st millennium BCE (Fig. 2). Analysing mtDNA, we found no significant genetic distance between those groups (Supplementary Table 8). We then used approximate Bayesian computation (ABC)[28] to test for continuity between the earlier (#4: Zevakino-Chilikta and #5: Aldy Bel; $n = 26$) and the later (#6: Pazyryk; $n = 71$) Scythian period in the East. The Tagar/Tes group (#7) had to be excluded because of their imprecise dating. These analyses revealed that they were most likely derived from one single population that was expanding over the time period considered here, that is, the two samples are unlikely to represent two independent populations that diverged earlier than 108 generations before present (g BP) or $\sim 2.7$ ky BP. This scenario was highly supported in our model selection procedure (Supplementary Table 9, logistic regression, $P = 0.995$, neural networks $P = 0.568$, cf. confidence in model choice in Supplementary Table 10, model parameter posteriors in Supplementary Table 11), whereas two scenarios that assumed that the eastern Scythian sample groups were derived from two previously diverged populations received very little statistical support (cumulative posterior probabilities: logistic regression, $P < 0.001$, neural networks, $P = 0.001$).

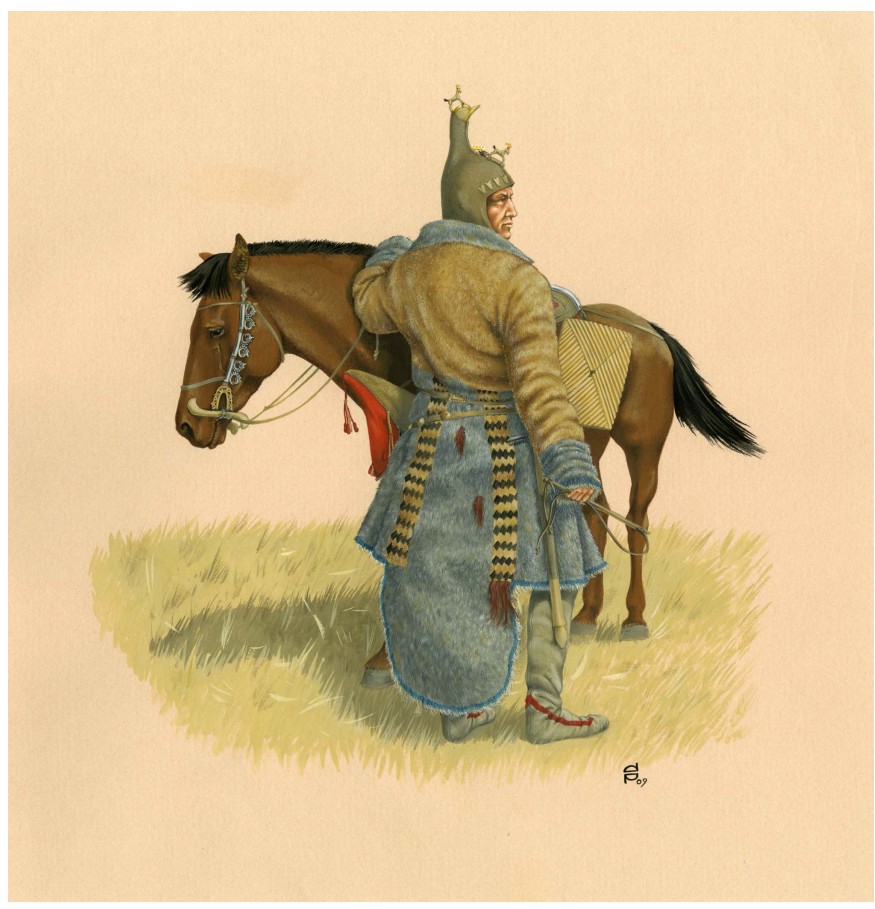

**Figure 1 | Reconstruction of a Scythian.** Found in the kurgan Olon-Kurin-Gol 10, Altai Mountains, Mongolia (reconstruction by Dimitri Pozdniakov).

Since the genetic distance between the combined Scythian groups of the east versus those of the west is relatively low ($F_{ST} = 0.01733$; $P$-value $= 0.02148 \pm 0.0045$), we used ABC to further assess if the eastern and western Scythians might share a common origin (Fig. 3). For these analyses we included contemporary samples representative of genetic diversity on the extremes of Eurasia (Supplementary Note 1). According to our model selection algorithm, a multiregional model provided the best fit to the empirically observed diversity patterns (Supplementary Table 12, 0.5% closest simulations, posterior probability $P = 0.708$ for logistic regression, $P = 0.715$ for neural networks method), while a model of western origin also received some support (Supplementary Table 12, 0.5% closest simulations, posterior probability $P = 0.286$ for logistic regression, $P = 0.267$ for neural networks method). Therefore, in contrast to the eastern origin model, a western origin cannot be fully discounted by our analysis. In addition, a pairwise comparison through the computation of Bayes factors reveals a substantial to strong (logistic regression) or weak to substantial (neural network) support for the multiregional origin over the western origin model (see Supplementary Note 1, Supplementary Table 14 and Supplementary Fig. 5b). These results suggest that western and eastern Scythian groups arose independently—perhaps in their respective geographic regions—and thereafter experienced significant population expansions (during the 1st millennium BCE). Importantly, our simulations support a continuous gene flow between the Iron Age Scythian groups, with indications of asymmetrical gene flow from western to eastern groups, rather than the reverse (see Supplementary Tables 13 and 15 for details).

Because population movements across Central Asia during the Bronze Age are often archaeologically associated with the spread of the Andronovo culture[29], we used ABC to fit a sample of Middle Bronze Age nomadic groups from western Siberia, most of them associated with the Andronovo culture, onto the preferred demographic model for the origin of Scythians. For this purpose—and based on low $F_{ST}$ values between these groups—we combined 40 samples related to the Andronovo culture in the west Siberian forest steppe[30] and nine samples from the same culture in the Krasnoyarsk region[31], all of which were dated to the first half of the 2nd millennium BCE. The results provided very strong support for a linkage between these Middle Bronze Age groups and eastern Scythians (Supplementary Tables 16 and 17). However, these simulations were not able to fully capture the patterns of genetic diversity observed in the Bronze Age populations, suggesting that the true demographic history of the ancestry of Iron Age populations may have been more complex than considered here (see Supplementary Note 1 and references 30 and 32 for details).

**Genetic diversity and ancestry of the Scythian groups.** Haplogroups found in the Iron Age nomads are predominant in modern populations in both west (HV, N1, J, T, U, K, W, I, X) and east Eurasia (A, C, D, F, G, M, Y, Z). The mitochondrial haplotype diversity in our sample set ranges from $0.958 \pm 0.036$ in the Tagar/Tes sample (group #7 in Fig. 2) up to $1.000 \pm 0.039$ in the Early Sarmatians (#3; Supplementary Table 7).

Using nuclear SNP data, we performed a principal component analysis[33] (PCA) of 777 present-day west Eurasians[26,34,35] onto which we projected the eight newly reported Iron Age Scythian samples as well as 167 other ancient samples from Europe, the Caucasus and Siberia from the literature[17,34,36] (Fig. 4). The two Early Sarmatian samples from the West (group #3 in Fig. 2) fall

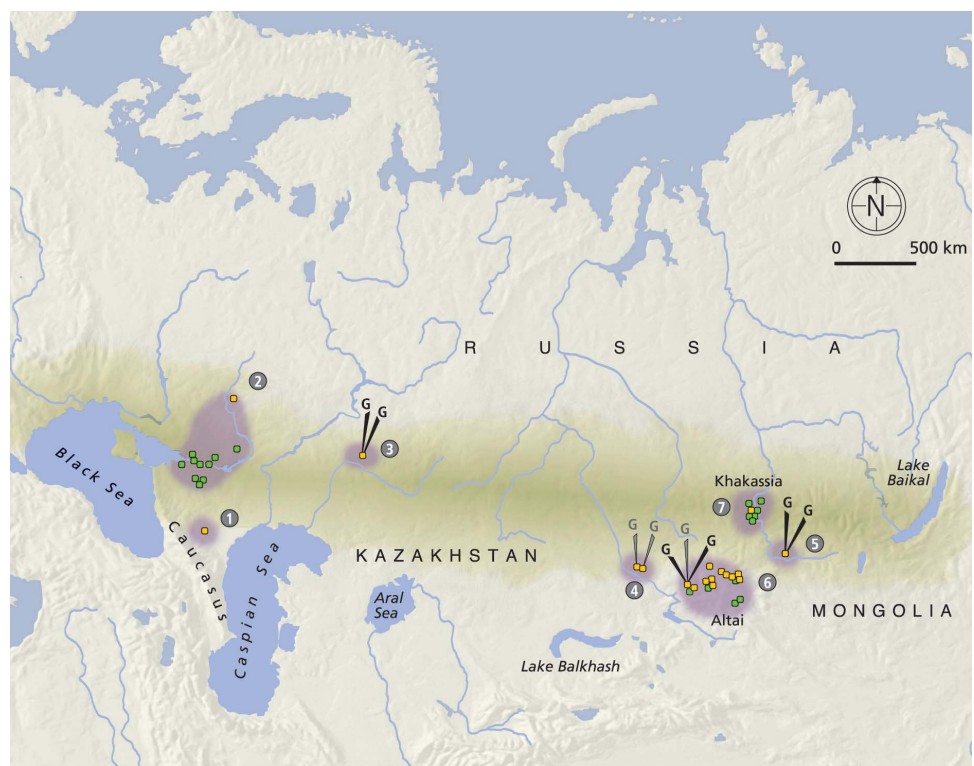

**Figure 2 | Distribution of the sample sites analysed for this study (yellow) including data from the literature (green).** Numbers refer to the defined groups (#): WEST: #1 initial Scythian period eighth to sixth century BCE (n = 4); #2 classic Scythian phase sixth to second century BCE (n = 19); #3 Early Sarmatians fifth to second century BCE (n = 11); EAST: #4 Zevakino-Chilikta phase ninth to seventh century BCE (n = 11); #5 Aldy Bel culture seventh to sixth century BCE (n = 15); #6 Pazyryk culture fourth to third century BCE (n = 71); #7 Tagar/Tes culture eighth century BCE—first century CE (n = 16); arrows with a G indicate samples for which genomic data was obtained, black for capture data and grey for shotgun data. Source of the map: cartomedia-Karlsruhe.

**Table 1 | List of all ancient individuals from which nuclear data were obtained in this study.**

| Sample | Site | Culture | Dating | No. SNPs overlapping the human origins array | Shotgun cov. Ø | Sample group from Fig. 2 |
|---|---|---|---|---|---|---|
| PR3 | Pokrovka, Russia | EarlySarmatian | 5th–2nd c. BCE | 306,498 | | 3 (West) |
| PR9 | Pokrovka, Russia | EarlySarmatian | 5th–2nd c. BCE | 186,890 | | 3 (West) |
| A10 | Arzhan, Russia | AldyBel | 7th–6th c. BCE | 427,557 | | 5 (East) |
| A17 | Arzhan, Russia | AldyBel | 7th–6th c. BCE | 108,952 | | 5 (East) |
| Be9 | Berel', Kazakhstan | Pazyryk | 4th–3rd c. BCE | 549,958 | 0.30 | 6 (East) |
| Be11 | Berel', Kazakhstan | Pazyryk | 4th–3rd c. BCE | 420,749 | | 6 (East) |
| Is2 | Ismailovo, Russia | Zevakino-Chilikta | 9th–7th c. BCE | 74,469 | 0.12 | 4 (East) |
| Ze6 | Zevakino, Russia | Zevakino-Chilikta | 9th–7th c. BCE | 163,338 | 0.28 | 4 (East) |

close to an Iron Age sample from the Samara district[34] and are generally close to the Early Bronze Age Yamnaya samples from Samara[26,34] and Kalmykia[17] and the Middle Bronze Age Poltavka samples from Samara[34]. The eastern samples from Pazyryk (#6), Aldy Bel (#5) and Zevakino-Chilikta (#4) are part of a loose cluster with other samples from Central Asia[17], including those from Okunevo, Late Bronze Age and Iron Age Russia, and Karasuk. These samples contrast with earlier samples from the Eurasian Steppe belonging to the Andronovo[17], Sintashta[17] and Srubnaya[34] groups, which overlap Late Neolithic/Bronze Age individuals from mainland Europe[34,35] and are shifted downwards in the PCA plot towards the early farmers of Europe and Anatolia[34].

Since the PCA of west Eurasia in Fig. 4 does not allow one to examine the ancient samples in relation to contemporary East Asian populations, we also carried out PCA of all 2,345 modern individuals in the Human Origins dataset[35], onto which we also projected the ancient individuals (Fig. 5). It is evident from this PCA that ancestry of the Iron Age samples falls on a continuum between present-day west Eurasians and eastern non-Africans, which is in concordance with the mitochondrial haplogroup analyses. The eastern Scythians display nearly equal proportions of mtDNA lineages common in east and west Eurasia, whereas in the western Scythian groups, the frequency of lineages now common in east Eurasia is generally lower, even reaching zero in four samples of the initial Scythian phase of the eight to sixth century BCE (group #1 in Fig. 2), and reaches 18–26% during later periods (sixth to second century BCE; #2 and #3) (Supplementary Table 7).

*f*-statistics. We used *f*4-statistics of the form *f*4(*Test*, LBK; EHG, Mbuti) and *f*4(*Test*, LBK; Han, Mbuti), which are zero for those *Test* samples that form a clade with LBK and positive for populations that have EHG- or Han-related ancestry, respectively. We

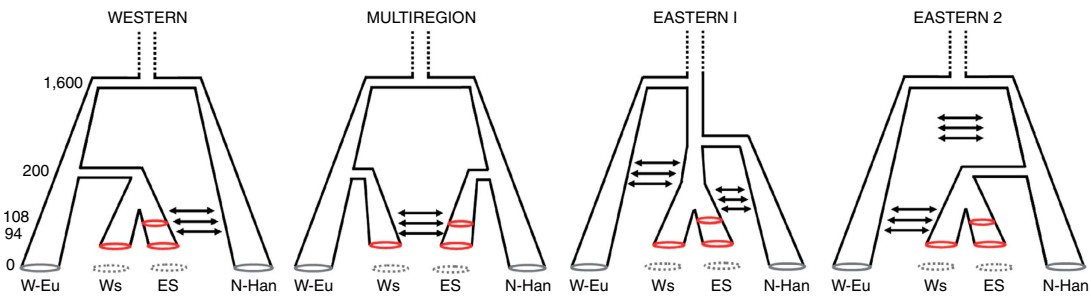

**Figure 3 | Candidate scenarios for the origin of Scythian populations.** W-Eu = West Eurasians; WS = western Scythian groups; ES = eastern Scythian groups; N-Han = Han Chinese. Numbers on the left refer to generations before present.

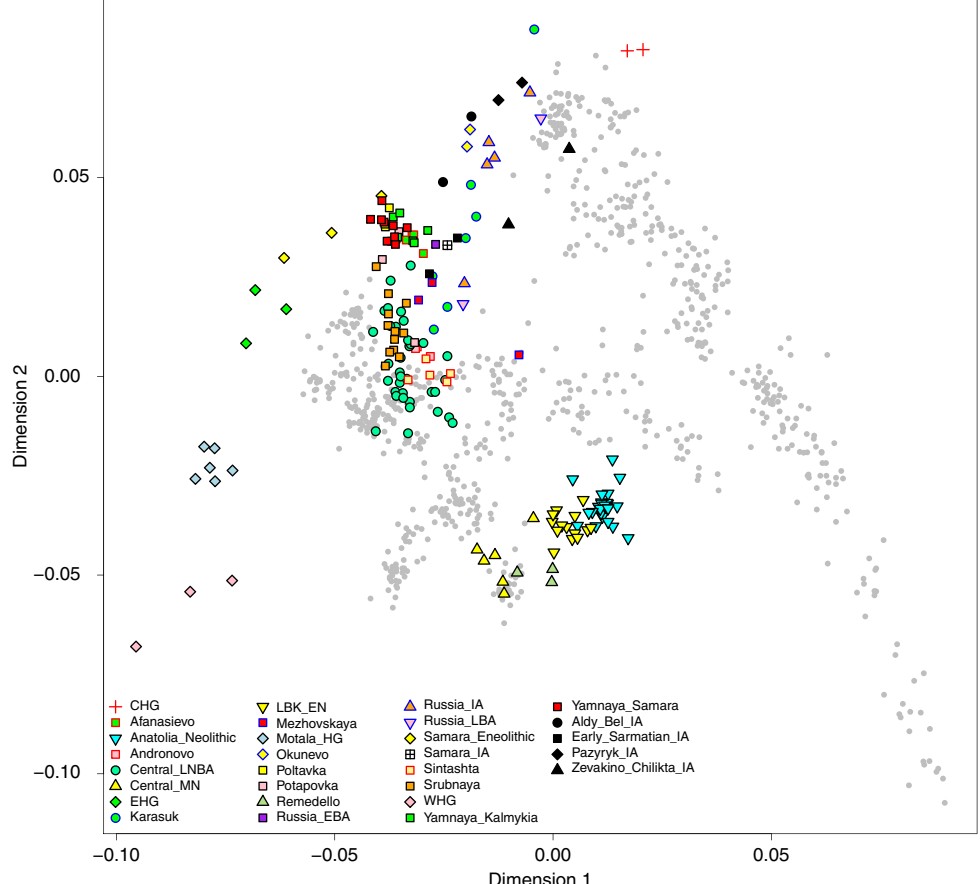

**Figure 4 | Principal component analysis.** PCA of ancient individuals (according colours see legend) projected on modern West Eurasians (grey). Iron Age Scythians are shown in black; CHG, Caucasus hunter-gatherer; LNBA, late Neolithic/Bronze Age; MN, middle Neolithic; EHG, eastern European hunter-gatherer; LBK_EN, early Neolithic Linearbandkeramik; HG, hunter-gatherer; EBA, early Bronze Age; IA, Iron Age; LBA, late Bronze Age; WHG, western hunter-gatherer.

plotted the results against each other, which resulted in a V-shaped pattern with Yamnaya at the apex (Fig. 6). The Iron Age Scythians are arrayed along a cline from Yamnaya to Ami (a population of East Asian ancestry that experienced no admixture), consistent with having ancestry from populations genetically similar to these two groups.

We also computed statistics of the form $f3(Test;$ Yamnaya_Samara, Han) to check whether a *Test* population has intermediate allele frequencies between Yamnaya_Samara and Han, which are used as proxies for possible source populations. Intermediate allele frequencies can only occur if the test population is a mixture of populations related to these

two sources[37]. These statistics are significantly negative for all Scythians, demonstrating that admixture occurred (Supplementary Fig. 13).

**ADMIXTURE analysis.** We carried out ADMIXTURE analysis[38,39] of 2,345 present-day humans[35] genotyped on the Human Origins array[35,37] and 175 ancient individuals on a set of 296,340 SNPs intersecting with those in the Human Origins array. The results for the ancient individuals are displayed in Fig. 7 for K = 15, which has the highest log likelihood value (the complete analysis can be found in Supplementary Fig. 14). All

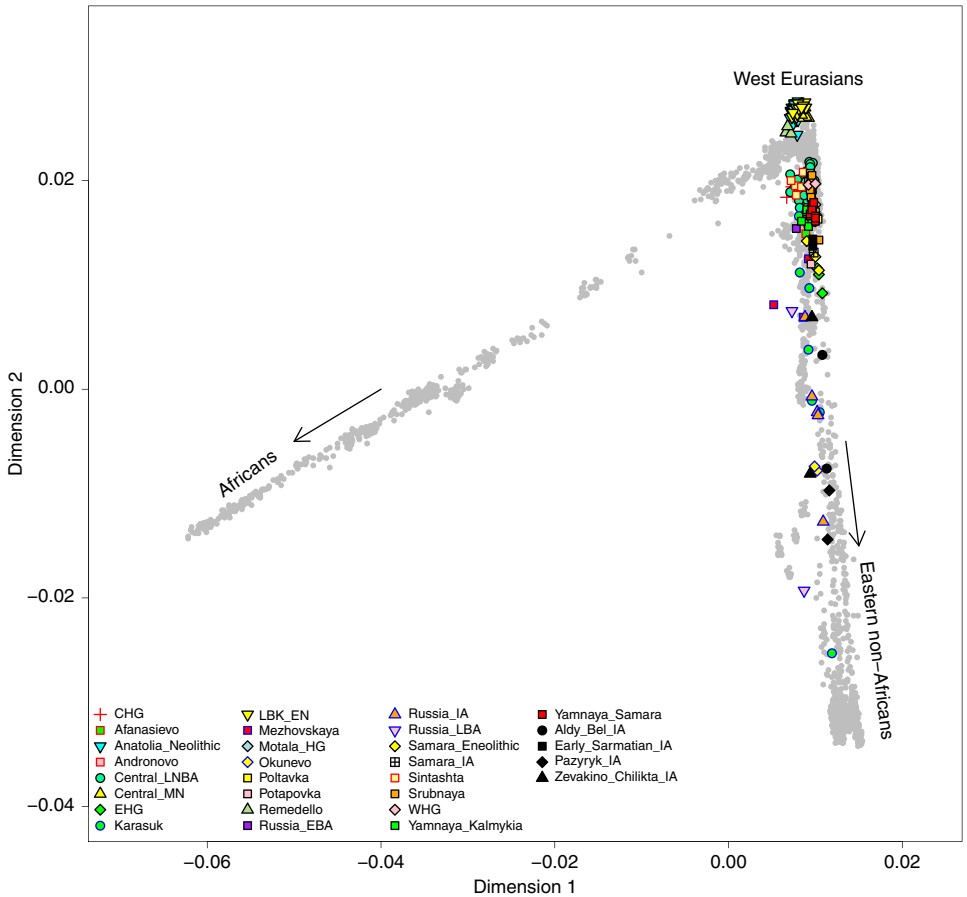

**Figure 5 | Principal component analysis.** PCA of ancient individuals (according colours see legend) projected on modern individuals of the Human Origins dataset (grey). Iron Age Scythians are shown in black; CHG, Caucasus hunter-gatherer; LNBA, late Neolithic/Bronze Age; MN, middle Neolithic; EHG, eastern European hunter-gatherer; LBK_EN, early Neolithic Linearbandkeramik; HG, hunter-gatherer; EBA, early Bronze Age; IA, Iron Age; LBA, late Bronze Age; WHG, western hunter-gatherer.

steppe populations have ancestry components that are maximized in European hunter-gatherers (blue) and Caucasus hunter-gatherers from Georgia[36] (green). One subset of the steppe populations (including Srubnaya, Sintashta and Andronovo) also have early farmer ancestry (orange), while a different subset (including all Iron Age samples) also have ancestry from a component (light blue) that is maximized in the Nganasan (Samoyedic people from north Siberian), and is pervasive across diverse present-day people from Siberia and Central Asia. Additionally, the Iron Age samples reveal an ancestral component that is maximized in East Asian populations (yellow), a type of ancestry that occurs at trace levels—if at all—among earlier steppe inhabitants, consistent with the observations from PCA and $f$-statistics about this type of admixture.

**Modelling ancient steppe populations.** We modelled steppe populations as mixtures of the Early Bronze Age Yamnaya and the LBK farmers from central Europe or East Asians (represented by the Han Chinese). We applied the method of *qpWave/qpAdm* used in Haak *et al.*[26], which provides a statistical test for the number of streams of ancestry into a *Test* population and allows one to estimate mixture proportions. In our application, we use five outgroups: Ust_Ishim[40], Kostenki14[41], MA1[42], Papuan and Onge. First we calculated whether *Test* and the Yamnaya from Samara could be descended from a single stream of ancestry. In the next step we included LBK farmers

testing whether *Test*, the Yamnaya from Samara and the LBK farmers from central Europe could be descended from two streams of ancestry, in which case *Test* could potentially be modelled as a mixture of the other two populations. Our results show that the Iron Age Scythians and the Yamnaya are not descended from a single stream of ancestry (Supplementary Table 23) and furthermore, cannot be modelled as mixtures of the Yamnaya and the LBK (Supplementary Table 24). We therefore considered an alternative model in which we treat them as a mix of Yamnaya and the Han (Supplementary Table 25). This model fits all of the Iron Age Scythian groups, consistent with these groups having ancestry related to East Asians not found in the other populations. Alternatively, the Iron Age Scythian groups can also be modelled as a mix of Yamnaya and the north Siberian Nganasan (Supplementary Note 2, Supplementary Table 26).

**Descendants of the Iron Age Scythians.** A multidimensional scaling (MDS) plot based on Reynolds' distances (Supplementary Fig. 1) suggests that the ancient Scythian populations from the eastern and western part of the Eurasian Steppe are genetically closer to each other than are the modern populations of the respective regions. AMOVA analyses carried out for modern and ancient groups of the eastern and western steppe provided further support for this finding. We found $F_{CT}$ values to be higher between modern populations of the East and the West ($F_{CT} = 0.0835$) than between ancient populations of similar regions ($F_{CT} = 0.0262$).

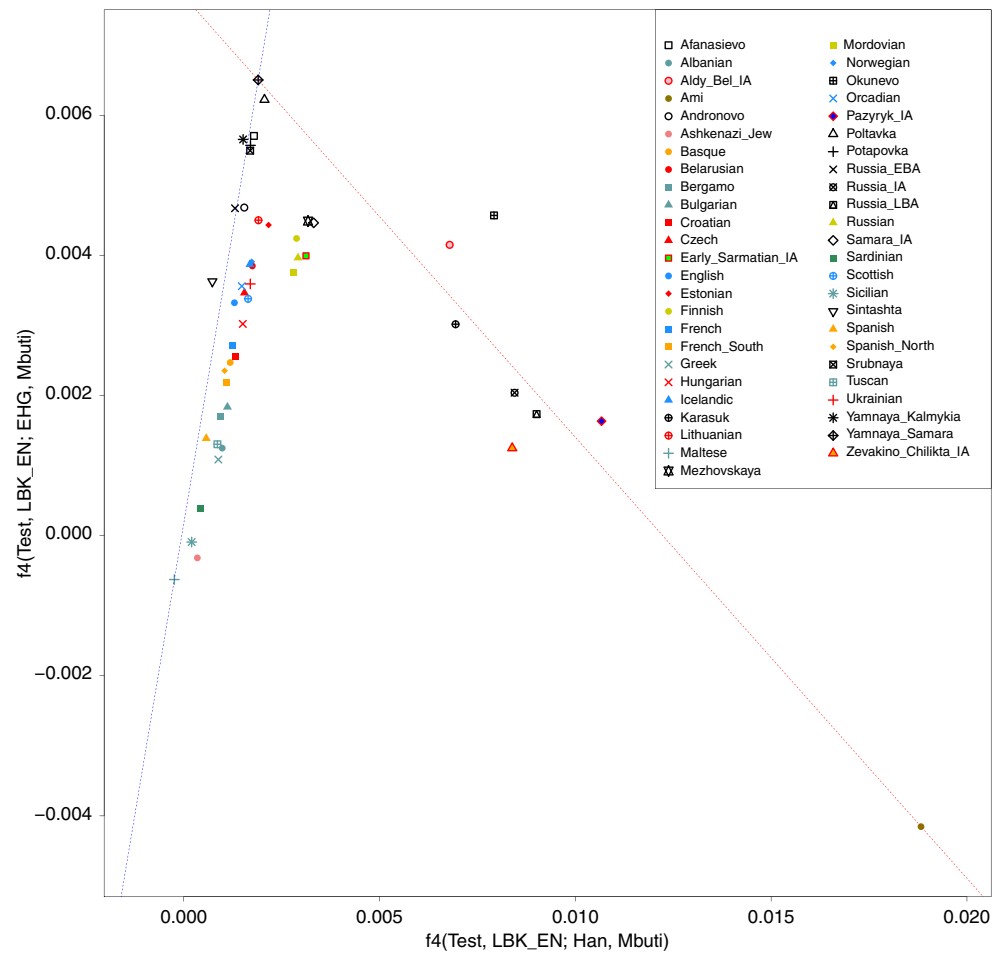

**Figure 6 | Visualization of *f*-statistics results.** *f*4(Test, LBK; Han, Mbuti) values are plotted on *x* axis and *f*4(Test, LBK; EHG, Mbuti) values on *y* axis, positive deviations from zero show deviations from a clade between Test and LBK. A red dashed line is drawn between Yamnaya from Samara and Ami. Iron Age populations that can be modelled as mixtures of Yamnaya and East Eurasians (like the Ami) are arrayed around this line and appear to be distinct from the main North/South European cline (blue) on the left of the *x* axis.

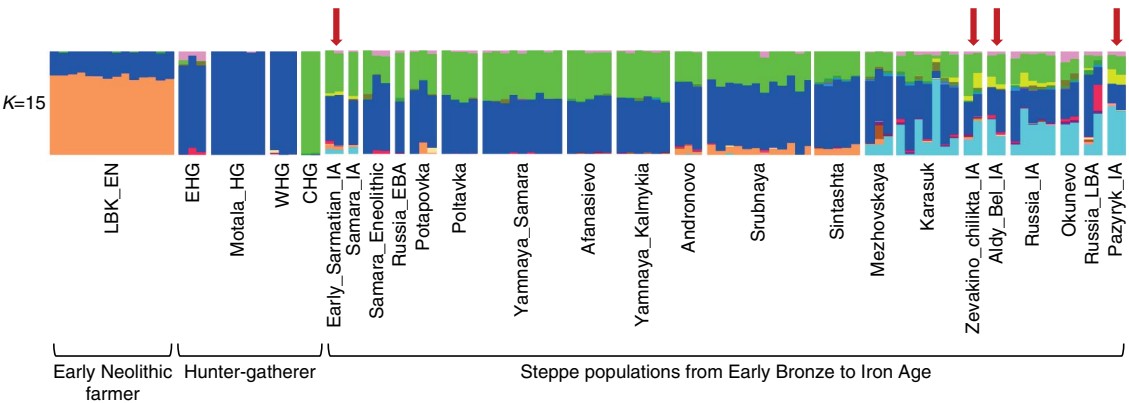

**Figure 7 | ADMIXTURE results for ancient populations.** Red arrows point to the Iron Age Scythian individuals studied. LBK_EN: Early Neolithic Linearbandkeramik; EHG: Eastern European hunter-gatherer; Motala_HG: hunter-gatherer from Motala (Sweden); WHG: western hunter-gatherer; CHG: Caucasus hunter-gatherer; IA: Iron Age; EBA: Early Bronze Age; LBA: Late Bronze Age.

A continuity test was performed between the two Iron Age groups ('West' and 'East') and a large set of contemporary Eurasian populations (*n* = 86, Supplementary Table 19). For western Scythian-era samples, contemporary populations with high statistical support for a genealogical link are located mainly in close geographical proximity, whereas contemporary groups

with high statistical support for descent from eastern Scythians are distributed over a wider geographical range. Contemporary populations linked to western Iron Age steppe people can be found among diverse ethnic groups in the Caucasus, Russia and Central Asia (spread across many Iranian and other Indo-European speaking groups), whereas populations with genetic

similarities to eastern Scythian groups are found almost exclusively among Turkic language speakers (Supplementary Figs 10 and 11).

**Phenotypic markers.** Derived alleles of pigmentation markers that are under selection in Europeans are present in eastern and western Scythians, including individuals who are homozygous for the derived alleles at selected SNPs in the *HERC2, SLC24A5* or *SLC45A2* (ref. 43–45). At the two *LCT* loci associated with lactase persistence, the derived allele is observed only in heterozygotes, only in the eastern Scythian samples, and at low frequency (2–3%). The ancestral alleles at *ADH1B* rs3811801 and rs1229984 are nearly fixed in the Scythian dataset, as they are in modern Europeans (the derived alleles, which confer some resistance to alcoholism, are under selection in East Asians[46,47]). We observe the derived allele at rs3827760 in the *EDAR* gene in a single Pazyryk individual (#6 in Fig. 2). This *EDAR* derived allele, which is related to tooth and hair morphology, is selected and at high frequency in modern East Asians (87%)[48], and very rare in modern Europeans (∼1%)[48], although it has been observed in prehistoric hunter-gatheres from Sweden (7.9–7.5 kya)[34]. Thus, the results of the examination of phenotypic SNPs that show frequency differences between Europe and East Asia are consistent with gene flow across the steppe territory.

## Discussion

Our results show that the Iron Age groups—long believed to be connected through shared cultural artefacts associated with the classical Scythians of the North Pontic region—also share a genetic connection. This is supported by our ABC analyses revealing population continuity over the 1st millennium BCE in the eastern Scythians and low $F_{ST}$ values between eastern and western Scythian groups. However, ABC analyses that evaluated different models for the origins of Scythian populations provided the strongest support for a multiregional origin, with eastern and western groups arising independently within their own regions. Despite separate origins and the enormous geographic separation, demographic modelling infers ongoing and substantial gene flow between eastern and western groups, which provides a plausible demographic mechanism to explain the low $F_{ST}$ values and the general uniformity of the material culture of Scythians right across the Eurasian Steppe zone.

Our genomic analyses reveal that western and eastern steppe inhabitants possess east Eurasian ancestry to varying degrees. In our ADMIXTURE analyses we find an East Asian ancestry component at K = 15 in all Iron Age samples that has not been detected in preceding Bronze Age populations in either western or eastern parts of the Eurasian Steppe. Another ancestral component that is maximized in the north Siberian Nganasan population becomes visible from the 2nd millennium BCE onwards in the eastern steppe (Okunevo, Karasuk, Mezhovskaya). This component appears later in all Iron Age populations but with significantly higher levels in the eastern steppe zone than in the West. These findings are consistent with the appearance of east Eurasian mitochondrial lineages in the western Scythians during the Iron Age, and imply gene-flow or migration over the Eurasian Steppe belt carrying East Asian/North Siberian ancestry from the East to the West as far as the Don-Volga region in southern Russia. In general, gene-flow between eastern and western Eurasia seems to have been more intense during the Iron Age than in modern times, which is congruent with the view of the Iron Age populations of the Eurasian Steppe being highly mobile semi-nomadic horse-riding groups.

In the East, we find a balanced mixture of mitochondrial lineages found today predominantly in west Eurasians, including

a significant proportion of prehistoric hunter-gatherer lineages, and lineages that are at high frequency in modern Central and East Asians already in the earliest Iron Age individuals dating to the ninth to seventh century BCE and an even earlier mtDNA sample from Bronze Age Mongolia[49]. Typical west Eurasian mtDNA lineages are also present in the Tarim Basin[16] and Kazakhstan[8] and were even predominant in the Krasnoyarsk area during the 2nd millennium BCE[31]. This pattern points to an admixture process between west and east Eurasian populations that began in earlier periods, certainly before the 1st millennium BCE[13,50], a finding consistent with a recent study suggesting the carriers of the Yamnaya culture are genetically indistinguishable from the Afanasievo culture peoples of the Altai-Sayan region. This further implies that carriers of the Yamnaya culture migrated not only into Europe[26] but also eastward, carrying west Eurasian genes—and potentially also Indo-European languages—to this region[17]. All of these observations provide evidence that the prevalent genetic pattern does not simply follow an isolation-by-distance model but involves significant gene flow over large distances.

All Iron Age individuals investigated in this study show genomic evidence for Caucasus hunter-gatherer and Eastern European hunter-gatherer ancestry. This is consistent with the idea that the blend of EHG and Caucasian elements in carriers of the Yamnaya culture was formed on the European steppe and exported into Central Asia and Siberia[26]. All of our analyses support the hypothesis that the genetic composition of the Scythians can best be described as a mixture of Yamnaya-related ancestry and East Asian/north Siberian elements.

Concerning the legacy of the Iron Age nomads, we find that modern human populations with a close genetic relationship to the Scythian groups are predominantly located in close geographic proximity to the sampled burial sites, suggesting a degree of population continuity through historical times. Contemporary descendants of western Scythian groups are found among various groups in the Caucasus and Central Asia, while similarities to eastern Scythian are found to be more widespread, but almost exclusively among Turkic language speaking (formerly) nomadic groups, particularly from the Kipchak branch of Turkic languages (Supplementary Note 1). The genealogical link between eastern Scythians and Turkic language speakers requires further investigation, particularly as the expansion of Turkic languages was thought to be much more recent—that is, sixth century CE onwards—and to have occurred through an elite expansion process. There are potentially many more demographic factors involved in the origins of Turkic language speakers, such as migration waves associated with Xiongnu, ancient Turkic or early Mongolian populations. The extent to which the eastern Scythians were involved in the early formation of Turkic speaking populations can be elucidated by future genomic studies on the historic periods following the Scythian times.

## Methods

**Sample material.** For this study, human skeletal sample material from different parts of Russia and Kazakhstan was selected based on an association of the archaeological complex with Scythian burial rites and artefacts. We analysed 110 skeletal samples. Five samples yielded no DNA, eight gave only poor results, and for one only coding region SNPs could be obtained. In the end we could use 96 samples for the analyses of mtDNA. Eight individuals were converted into Illumina libraries; six thereof were used for a genomic capture, two for shotgun sequencing, and one for both capture and shotgun sequencing. Overall DNA preservation was remarkably good, with slight variations depending on region or sample site (Supplementary Tables 1 and 2).

**Sample preparation.** All pre-PCR sample preparation steps were carried out in a cleanroom facility physically separated from the post-PCR laboratories. Sample preparation, DNA extraction, the amplification of single mtDNA fragments and

the amplification of nuclear loci were performed as previously described[45]. For mtDNA analyses 30 coding region fragments covering 32 haplogroup-specific SNPs of the mitochondrial genome were selected from the literature[51–53]. Primer systems (Supplementary Table 3) were designed using PrimerSelect, a part of the Lasergene software package (DNASTAR). Amplification was performed in three different multiplex reactions to avoid overlapping fragments (Supplementary Table 3). The Multiplex PCRs were carried out in a total volume of 40 μl, using 20 μl Qiagen Multiplex Kit Master Mix (Qiagen, Hilden, Germany), 11 μl ddH$_2$O, 0.2 μl of each primer and 4 μl DNA extract. Multiplex PCR conditions were as follows: initial denaturation for 15 min at 95 °C followed by 36 cycles of 40 s denaturation at 94 °C, 40–90 s annealing at 56 °C, 40–90 s elongation at 72 °C and for some reactions a final elongation for 10 min at 72 °C was added. Standard PCR was used for fragments that failed amplification in the multiplex reactions and to amplify the HVR1 of some additional samples.

**Sequence analyses and authentication.** To allow for parallel sequencing of 37 mtDNA fragments in up to 60 samples on a 454 GS FLX machine, barcodes were ligated to the multiplexed DNA fragments following the tagging protocol of Meyer 2008 (ref. 54), with the exception that the whole PCR product was used for the initial step. The three multiplex PCRs of one sample were pooled during the first purification step. For all purification steps the Qiagen MinElute PCR purification Kit was used. 454 library preparation and sequencing was carried out by GATC Biotec AG in Konstanz. Single PCR products were analysed using the ABI 3130 Sequence Analyzer after standard Sanger-cycle-sequencing using BigDye Terminator v1.1 Cycle Sequencing Kit (Applied Biosystems, Life Technologies, Darmstadt, Germany). For the nuclear markers redundant sequences were collapsed using CD-HIT-454 (ref. 55), single sequence clusters were removed and unique sequences were aligned with mafft[56] to a reference sequence for each of the markers and alleles were counted. For mtDNA sequences consensus sequences were created with SeqMan of the DNASTAR Lasergene package, and haplogroups were assigned with HaploGrep, a web-based application using phylotree build 15 (refs 57,58) (Supplementary Tables 4–6). To authenticate the results, every position of the HVR1 had to be covered by at least three unambiguous sequences from independent PCRs. The coding region fragments were sequenced at least twice with the 454 FLX. Independent reproduction has been carried out for 22 of the samples by A. Pilipenko at the Institute of Cytology and Genetics, Siberian Branch, Russian Academy of Science. All results were concordant with results produced in the Palaeogenetics Lab, Mainz. During the sample preparation process blank controls were included in the pulverization, extraction and amplification steps. Overall contamination rate was 2.2%. The contaminations were monitored and compared with sample sequences of the same reaction steps. No correlation could be detected and since the sample sequences were reproduced in independent reactions all results are considered authentic.

**Biostatistical analyses.** For the population genetic analyses HVR1 sequences from position 16,040–16,400 of 96 samples analysed for this study and additional 51 samples taken from the literature (Supplementary Table 1) were used. The Arlequin 3.5.1.3 software[59] was used for AMOVA and to calculate gene diversity (haplotype diversity)[60], nucleotide diversity[60,61], $F_{ST}$ values and Reynolds' distance[62]. The pairwise distance method was used to calculate the $F_{ST}$ values with 1,000 permutations and a gamma value = 0. Fu's $F_S$ test of selective neutrality was performed to test for population expansion. The significance level for the $F_S$ values was set at 0.02 (ref. 63). For results of summary statistics see Supplementary Table 7.

**ABC analyses.** To explore the demographic history of Scythians we formulated multiple candidate scenarios, which provided the basis for simulating samples of genetic data for the HVR-1 region, using BayeSSC[64,65]. Calculations of summary statistics for the observed data were performed in DNaSP v5 (ref. 66). To confirm that candidate scenarios were able to reproduce the observed genetic data, we compared the prior distributions of simulated summary statistics with the empirically observed values. All analyses were performed using the *abc* package in R 2.15.1 (refs 67,68). We refer to the Supplementary Note 1 for full details on demographic scenarios and analyses employed here.

**Genomic analyses.** DNA-library preparation for subsequent shotgun sequencing was performed according to the protocol used in Kircher 2012 (ref. 69) with slight modifications for the shotgun samples (Supplementary Note 3). To analyse the genomic data $f_3$- and $f_4$-statistics were calculated (qp3Pop and qpDstat from ADMIXTOOLS)[37], analysis of ancestry streams was applied (*qpWave/qpAdm*)[26] and an ADMIXTURE analysis was performed[38,39] (Supplementary Note 2).

**Data availability.** Bam files for the genomic data can be downloaded from the European Nucleotide Archive under accession number PRJEB18686. Mitochondrial sequences were deposited in GenBank under accession numbers KY369766–KY369861.

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

## Acknowledgements

We would like to thank Leonid Yablonsky for providing the Pokrovka samples and Jeannine Davis-Kimball for her help during the sampling process; Maria Mednikova and Maria Dobrovolskaya for the provision of the samples from Novozavedennoe and Kolbino, respectively; Anton Gass and Anatoli Nagler for additional archaeological information. Some computationally intensive analyses were run on the Linux cluster of the 'Museum National d'Histoire Naturelle' in Paris (administrated by Julio Pedraza Acosta). D.R. was supported by NIH grant GM100233, by NSF HOMINID BCS-1032255, and is a Howard Hughes Medical Institute investigator. This work was sponsored by the German Federal Ministry of Education and Research. M.C. was supported by Swiss NSF grant 31003A_156853. Ancient DNA experiments in the Novosibirsk lab were financed by a Russian Science Foundation (RSCF) grant (project No. 14-50-00036).

## Author contributions

J. Burger and H.P. designed the study; E.K., W.S., D.P., A.K., H.P., Z.S. and V.I.M. provided sample material, archaeological background information and/or anthropological information; M.U. and A. Pilipenko processed aDNA samples and provided mitochondrial sequences; M.Groß, J. Blöcher processed ancient DNA samples and prepared NGS libraries; Z.H., M. Groß, C.S., J. Blöcher and M.C. analysed shotgun data; I.L., N.R. and D.R. performed nuclear capture and subsequent genomic analyses on SNP data; F.P. designed and performed ABC analyses on mtDNA; M.U., Z.H., F.P., A. Powell, M.C., H.P. and J.B. interpreted the results and put them in context; M.U., K.K. and S.W. processed phenotypic SNP analyses on 454 data; B.R. provided code for 454 data analyses; M. Georges and E.H. analysed and provided modern mtDNA data from central Asia; M.U., F.P., I.L., K.K., D.R. and J. Burger wrote the paper with help from co-authors.

## Additional information

**Competing financial interests:** The authors declare no competing financial interests.

**Publisher's note**: 

