## [Peer Review File · Nature Communications]

Reviewers' comments:

Reviewer #1 [Remarks to the Authors]:

In this manuscript, the authors present an ancient DNA dataset and interpret the results in terms of demographic events related primarily to the Scythian culture. This is an interesting if eclectic dataset, and the sampling of individuals of an ancient culture across both geography and time makes it fairly unique.

The authors have performed an impressive amount of analysis on the new data. As far as I can tell, most of it was done competently. However, I found the interpretation very difficult to follow. It was especially difficult to distinguish between meaningful findings and less significant (or even statistically insignificant) tangential observations. By contrast, it often felt like key arguments were dispersed between the results, discussion, methods, and supplementary sections, making for a painful read.

It may be that the breadth of material covered would require a longer format. My reaction may be heavily influenced by my relative ignorance of early Eurasian settlement. I assume that most readers of this journal will be in a similar situation, however, so it may be worth thinking about making the material more accessible -- some suggestions below.

This is my understanding of the most important results that have been shown:

- 1-There is population structure in the sample,
- 2-There is continuity between the ancient and modern populations leading to isolation-by-distance that is maintained across time
- 3-Isolation by distance is not uniform, suggesting that migrations were also not uniform.
- 4-There was enough migration over time so that population trees with isolated branches do not explain the data.
The best-fit model is a multiregional model with independent appearance of Scythian populations followed by migration. (However, I have a some concerns about this result.)
- 5-Earlier populations from the Andronovo culture appear more closely related to the Eastern Scythian branch.

1-The most important claim of the paper, I believe, is the multiregional origin. The authors make a strong claim:
"ABC analyses that considered different models for the origins of Scythian populations during the Iron Age provided strong support for
a multiregional origin, with eastern and western groups arising independently within their own region. There is no support for a scenario of a single origin of Scythians in either the East or the West followed by a migration wave in one direction or the other"

However, looking at Table S14, it seems like the posterior probabilities for the Western origin model are comparable to, and for some parameter choices even higher than, those of the multiregional model. In the most favorable parameter choice, the posterior for the multiregional model is only 0.76 compared to 0.24 for the Western origin model.

As I understand things, the correct interpretation is that the multiregional model is only slightly preferred, and that there is insufficient power to distinguish between the models.

2-

Even though the writing is quite clear at the sentence level, the general presentation of the results made it a very difficult read for someone with my background.

Most results depend on quantitative analysis and relatively complex demographic scenarios over multiple places and time periods, but there is not a single figure in the main text illustrating the models used, the analysis strategy, or the results.

Even though the artistic depiction of a Scythian on Figure 1 provides an image of what the population may have looked like, it would have been much more useful for me to see

- an illustration of the main demographic scenarios used,
- a more detailed summary of the different cultures, populations, and sampling locations
- some of the results

Perhaps because of this lack of visual guidance, I found that the results and discussion read like a list of loosely related observations, without a clear thread.

3-

Consider, as a representative example, the middle paragraph on page 11. It contains a description of a number of phenotype-associated SNPs and their observed frequency in the different samples. Unfortunately, because of the small number of loci and samples, it is impossible to draw a statistically meaningful conclusion from these data. The authors conclude by stating that these results are consistent with on-going gene flow across the steppe territory. This is not untrue, but 1- this conclusion has nothing to do with the phenotypes, and 2- as far as I can tell, the data are also consistent with the absence of on-going gene flow across the steppe territory.

Unless I missed something, this paragraph neither supports nor confirms the main hypotheses. Many paragraphs similarly read like observations only loosely connected to the previous or following observations.

4-

Another style issue is that many of the genetic analysis discussions that are provided in the main text are not self-contained.

For example, the second paragraph on page 5 reports the results of an f_4 statistic analysis. However, there is no discussion of what was the null model, and the reader has to go to the supplement to learn that the test that was done was $f_4(\text{Yamnaya, Scythian; Test, Chimp})$ for different "test" populations.

It's already demanding of the reader to understand that $f_4(\text{Yamnaya, Scythian; Test, Chimp})$ means:

1- There is a null model in which Yamnaya and Scythian form a clade relative to populations "test" and chimp.

2- The z-score represents the departure from this assumption as measured by excess similarity between "test" and one of Yamnaya and Scythian.

But the main text does not give an indication of which clades are assumed in the null model. It doesn't even mention the Yamnaya! It's fine to leave some statistical detail to the supplement, but I believe that the reader must be provided with an idea of what is being tested.

This is particularly problematic because the results of the test are not always presented in quantitatively meaningful terms: the f_4 test shows that the "East Eurasians are more closely related to the Scythian group than to the Yamnaya", however, the authors conclude that "The Scythian group

has a strong genetic component that resembles current East Asian populations". I am not sure of exactly what the authors mean by "a strong genetic component", but I haven't seen evidence for what I think it means.

4- Another example of the same problem, on page 7:

" We tested for continuity between the earlier (Zevakino-Chilikta and Aldy Bel; n=26) and the later (Pazyryk; n=71) Scythian period in the East using approximate Bayesian computation (ABC). These analyses revealed that they were most likely derived from one single population that was expanding over the time period considered here."

There is not enough information in the main text to understand what was done, and no reference for the reader as to where the relevant information might be.

I find it surprising to find such strong support for a single origin model without any support for the split model, given that the single-population model should be nested within a population split model.

In other words, there should not be sufficient resolution to distinguish a relatively recent split between the populations. The fact that the manuscript conclusively rules out a split suggests that either the correct split model was not included in the tested models, or that the prior probabilities unfairly penalized the more complex model.

5-

It is hard for a non-specialist to jump back and forth between names of populations/cultures, time periods, sampling locations, sample ID, and modes of subsistence. This was a strong barrier for me throughout the manuscript.

For example, in the Admixture discussion (Figure S12), the text refers to the "ancient steppes genomes", the Scythian individuals, the European Hunter-Gatherers, and so forth, but Fig S12 uses different labels. I figured it all out eventually, but that required energy that could have been better spent elsewhere.

6-

The authors face a very challenging sampling problem since they need to be representative in both space and time. Many estimates are based on very few genomes which may or may not be representative of broader populations. These estimates are often provided without a mention of statistical uncertainty.

For example, the first paragraph of page 10 looks at a time course of ancestry proportions over time; it would have been interesting to discuss whether these changes are statistically significant.

Similarly, the proportion of "eastern" mitochondrial lineages identified in the Aldy Bel sample is listed in the main text as "66.7%" (p.6) , and this is compared, e.g., to the proportion in modern Altaians of "65.5%". Looking through the supplement, the proportion is based on 15, so 10 out of 15 samples had an "eastern" haplotype. Reporting three significant figures for such a small sample size is misleading.

7. The analysis of the mtDNA data makes use of and MDS figure and FST, and FCT, but not of an inferred tree, which seems to be the natural way to display mtDNA sequence relatedness. Why is that?

8-The admixture analysis using Shotgun sequencing is unconvincing (Figs S27-S29.) For example, Figure S29 shows Europeans having about 60% of the East Asian component, against 5% in S28 and 15% in S27. Given these large differences for the same individuals in different runs, why should we even bother to compare values inferred for different ancient individuals across runs?

9-Some of the results are split between the results and the discussion section with no obvious rationale.

For example, ABC simulations supporting continuity are presented in the results section, whereas the results supporting the mutiregional origin are presented in the discussion. This is particularly confusing because the multiregional model, presented on page 11, is necessary to understand the Andronovo discussion, presented on page 8.

Reviewer #2 [Remarks to the Authors]:

This work describes genome wide data from 8 and mtDNA sequence from 96 Eurasian Steppe individuals and ties a demographic model of origins to a specific cultural complex. My overall opinion is that within these data and their analysis there is a basis for publication in Nature Communications. However, I have criticisms of the work as it is presented which need to be addressed.

I found it difficult to follow the results which support the inferences made in this paper, chiefly the different ancestries of east and west Scythian nomads. The text argument really must be better supported, paragraph by paragraph, by figures and tables integrated into the main text - as is allowed by this journal. Currently there are only two figures, neither of which convey genetic data or analysis! Some recommendations below:

ADMIXTURE. This is the first analysis mentioned in the discussion and one of the main ways of conveying structure in autosomal data - e.g. it has been used effectively in Allentoft et al. Here, it should be displayed, focusing on the most defensible value of k (9 perhaps). A wider analysis should be considered, are there other Iron Age genome wide samples that could be incorporated from prior data - I recall Allentoft et al, Gamba et al, perhaps Mathieson et al may have such. The mtDNA results incorporate data from the literature, but the autosomal analyses do not. Can the shotgun and capture data used here be combined into a single digram, perhaps using an ANGSD based analysis?

Shotgun coverage levels are a key part of the work and should be quoted in the main text, not referred to in a remote supplementary table.

PCA - a common plot for all ancient genome wide data should be shown in the main text.

f_4 , f_3 statistics. Can these be displayed graphically in the main text. Perhaps incorporated with a map - it is difficult for a reader unfamiliar with the archaeology to translate the populations into positions in space and time as s/he reads this. Perhaps a summary table of key tests and findings, e.g. as used by Allentoft et al.

A figurative display of the mtDNA modelling results needs to be incorporated into and defended in the main text. These are key to the papers argument. Perhaps a version of the map in supp fig 11 and a cartoon of the key models used.

Reviewers' comments:

Reviewer #1 (Remarks to the Author):

The authors have made substantial improvements to the manuscript, especially regarding legibility. I still have some concerns, but I trust that they can be addressed fairly easily.

1-The authors have not satisfactorily addressed my concern about the degree of confidence in the multiregional model. The western origin model, whose posterior probability is 24%, cannot be considered "rejected" as the authors do on line 293. The argument made in the rebuttal, that the ABC has high power in simulations, is entirely unconvincing.

If the data provides similar support to two models, one cannot just consider the one with most support as strongly supported just because the same method applied to a different dataset has provided a high confidence call.

When one simulates according to the model used in inference, one naturally has higher power to identify the correct model. In fact, I suspect that the posteriors for the correct model in the simulations were stronger than the support for the multiregional model in the data.

Finally, even in the idealized context where simulations and inference model use the same model, the evidence appears weak. According to table S14, the Western Origin model would get rejected 11% of the times, and 9% in favor of the multiregion model, even if it were true. This is not even marginal significance for rejecting a model.

The data shows only weak support for the multiregional model. The easy way to address this concern is to water down the claims accordingly throughout the manuscript. If the authors want to argue that there is strong support, then they need to provide a more compelling argument.

2-lines 120-130

Similarly, the argument for continuity should be clarified. The additional simulation performed does add support to the proposed interpretation. However, the main text should specify that the split model only considered splits older than 108 generations ago. The evidence is that the two populations are unlikely to have diverged more than 108 generations (or 2.7ky) ago. This is useful evidence, but more specific than what is claimed in the main text.

This section should also point the reader to the supplement for the detailed information.

3- I206: why does being shifted away from Yamnaya indicate Yamnaya ancestry?

Minor points and typos, etc.

Throughout: Principal components, not principle components

Reviewer #2 (Remarks to the Author):

The paper has been substantially altered, with more argument centred on the data and analysis in the main text.

I have some remaining criticisms.

line 129 p value of 0.000 doesn't make sense, surely $p < 0.001$

line 131-141 This paragraph is central as it describes the conclusion of a multi regional model for origins. The posterior probability of the favoured model is given as ~ 0.7 . however, the reader is not furnished with the evidence selecting this model rather than the other options. Is it significantly more likely? If so, what is the weighting in favour of it?

line 202-208 The interpretation of the f_4 test is not made very clear. What is the line in figure 6 meant to signify? Why does a shift to zero (which above is taken to indicate LBK ancestry) indicate Yamnaya ancestry - line 206. Perhaps labelling key groups of populations on the plot itself might help the reader to access this information better, it is difficult from the key alone.

line 209 $f_3(\text{Test}; \text{Yamnaya_Samara}, \text{Han})$ to check whether a Test population has intermediate allele frequencies between Yamnaya_Samara and Han, which can only occur if it is a mixture of populations related to these two sources.

Is this strictly true? What about an ancestral population. Surely these populations need not be the actual sources, but proxies related to sources.

REVIEWERS' COMMENTS:

Reviewer #1 (Remarks to the Author):

The modifications to the manuscript are a step in the right direction, yet I still have a number of concerns. Once again, these concerns do not necessarily require new analyses, but are requests for clarifications and more measured conclusions.

1-

I am still concerned about model selection and the discussion surrounding the multiregional vs western origin model. The manuscript reinterprets the same signal through through many different model selection tools. It's fine to report all the tests to show consistency, but it should be made clear that these are not independent lines of evidence.

For example, after acknowledging that the posteriors do not offer definitive evidence in support of one model versus the other, the manuscript try to strengten the claim (l. 142):

"However, a pairwise comparison through the computation of Bayes factors reveals a strong (logistic regression) or substantial (neural network) support for the multiregional origin over the western origin model (see Supplementary Information 1, Supplementary Table 14, Supplementary Figure 5b). "

This is not additional evidence: All these methods are testing for the same signal in slightly different ways. This sentence appears to suggests that we should pay special attention to the last two Bayes factor tests. Given that there were over 40 tests performed on Figure S5b, there should be a correction for multiple testing. If the different tests are just shown for consistency, the authors can't cherry-pick the most significant tests.

Second, the claim that support is "strong" for logistic regression and "substantial" for neural network is not supported by figure 5b: Most tests using the Neural network provide "weak" evidence (rather than "substantial"), and most tests with the logistic regression offer "substantial" support (rather than "strong"). This is reported correctly in the supplement, but not in the main text.

Third, it may be useful to provide K values together with the interpretation "weak/substantial/strong" scale. As I understand things, "strong" support would be the bar typically used to establish statistical significance (corresponding roughly to a $p=0.05$ in a frequentist approach). The additional Bayes factor analysis therefore basically repeats what had been observed already: the multiregional model has a somewhat higher posterior.

2-

The supplement states that:

"These inferences become more pronounced when repeating analyses with only the western and multiregional candidate models (multiregional model; 0.5% closest simulations, posterior probability $p=0.917$ for logistic regression, $p=0.929$ for neural networks method)".

Shouldn't the posteriors of the two remaining models be in proportion to their posterior probabilities when three models were considered?

3-

L 306: "However, a common origin for all of these groups either in the West or the East is unlikely (western origin) ..."

This is not supported by the analysis. I don't think that a 27% posterior probability makes a model

"unlikely". Given the weak signal, it is also quite plausible that peculiarities of the model choice tipped the balance in favor of the multiregional model. For example, results might have been very different if direct migration were allowed from Asia into the WS in the western origin model, or gene flow between the WS and ES. It's never possible to test all models, and the manuscript does explore a reasonable set of models, but the supported conclusion is that the multiregional model was the best fit, not that a western origin is unlikely.

4-

"Importantly, gene flow between the Iron Age Scythian groups was ongoing and substantial, with asymmetrical gene flow from western to eastern groups, rather than the reverse (see Supplementary Table 17 for details)" (l. 147)

and

"Despite separate origins and the enormous geographic separation, demographic modelling infers ongoing and substantial gene flow between eastern and western groups" (L 309")

I think that the authors mean Table S15, since I do not see that Table S17 is related to this discussion. Looking at Table S15, I do not see statistical support for asymmetric gene flow (since the confidence intervals overlap substantially).

I did not see units provided for the migration rate, so it's hard to judge about whether migration is substantial.

Finally, where was "ongoing" migration shown? I assume that the alternative is punctuated gene flow, but where was this ruled out?

5-

"The two Early Sarmatian samples from the West (group #3 in Fig. 2) cluster with an Iron Age sample from the Samara district and are generally close to the Early Bronze Age Yamnaya samples from Samara and Kalmykia and the Middle Bronze Age Poltavka samples from Samara. The eastern samples from Pazyryk (#6), Aldy Bel (#5), and Zevakino-Chilikta (#4) are part of a loose cluster with other samples from Central Asia, including those from Okunevo, Late Bronze Age and Iron Age Russia, and Karasuk."

"Cluster with" is an overstatement. It also seems to me that the Karasuk are closer to the Early Sarmatian than are groups #4-6.

It would be useful to have a main conclusion to this paragraph.

6-

"Since the PCA of west Eurasia in Figure 4 does not allow one to examine the relationship of the ancient samples to contemporary world populations, we also carried out principal component analysis of all 2,345 modern individuals in the Human Origins dataset,"

I do not understand why "the PCA of West Eurasia in Figure 4 does not allow one to examine relationships [to present-day populations]". After all, this was performed by projecting the ancient samples onto PCAs formed by present-day populations. Wouldn't it simply be a matter of coloring the gray dots? I don't see how adding Africans to the PCA helps resolve anything about the ancient samples. It would only hide the finer-scale relationships, wouldn't it?

7-

L 217 "The Iron Age Scythians are arrayed along a cline from Yamnaya to Ami (a population of East Asian ancestry that experienced no admixture), consistent with having ancestry from populations genetically similar to these two groups."

L 228 "Iron Age populations that can be modelled as mixtures of Yamnaya and East Eurasians"

Unless I am missing something, this is a very weak argument: There is no obvious cline along the red line, nor is there obvious "arraying". I see a line that was drawn between two arbitrarily chosen points, and that happens to fall close to another set of points.

It's also unclear to me whether a mixture population would indeed fall halfway between the source populations on this graph.

Figure 6 does suggest that the Iron Age populations are somewhat more closely related to the Asian populations than the European populations, but this is a strange way to go about showing this. I don't think that it adds information beyond what is already visible from Figure 5.

8-

I.297: "Thus, the results of the examination of phenotypic SNPs that show frequency differences between Europe and East Asia are consistent with on-going gene flow across the steppe territory."

I'm not sure how this conclusion is reached. I agree that the observations are not inconsistent with on-going gene flow, but they do not really support gene flow, let alone "ongoing" gene flow. Why is the consistency of phenotypic SNPs to an ongoing gene flow model relevant?

9-

L322: "In general, gene-flow between eastern and western Eurasia seems to have been more intense during the Iron Age than in modern times"

This sounds plausible, but I do not see where this was discussed.

More minor points

10-

I. 172: "We find very high mitochondrial haplotype diversity in our sample set, ranging from 0.958 ± 0.036 in the Tagar/Tes sample (group #7 in Fig. 2) up to 1.000 ± 0.039 in the Early Sarmatians

Very high with respect to what?

11-

"These samples contrast with earlier samples from the Eurasian steppe belonging to the Andronovo, Sintashta and Srubnaya groups, which overlap Late Neolithic/Bronze Age individuals from mainland Europe and are shifted 'southwards' in the PCA plot towards the early farmers of Europe and Anatolia."

Is "southwards" used to refer to the bottom of the PCA figure? If so, using geographical terms for this purpose is confusing. If it refers to geography, please explain the mapping.

12-

Figure 4 has no axes labels.

13-

Supplementary Table 11 does not appear to be quoted in the text.

Reviewer #2 (Remarks to the Author):

My prior concerns are now dealt with adequately

24.08.2016

Dear Editor,

Please find below our point-by-point response to the referees. We repeat the reviewers' comments in black, and give our response in purple:

Reviewers' comments:

Reviewer #1 [Remarks to the Authors]:

In this manuscript, the authors present an ancient DNA dataset and interpret the results in terms of demographic events related primarily to the Scythian culture. This is an interesting if eclectic dataset, and the sampling of individuals of an ancient culture across both geography and time makes it fairly unique.

The authors have performed an impressive amount of analysis on the new data. As far as I can tell, most of it was done competently. However, I found the interpretation very difficult to follow. It was especially difficult to distinguish between meaningful findings and less significant (or even statistically insignificant) tangential observations. By contrast, it often felt like key arguments were dispersed between the results, discussion, methods, and supplementary sections, making for a painful read.

It may be that the breadth of material covered would require a longer format. My reaction may be heavily influenced by my relative ignorance of early Eurasian settlement. I assume that most readers of this journal will be in a similar situation, however, so it may be worth thinking about making the material more accessible -- some suggestions below.

This is my understanding of the most important results that have been shown:

- 1-There is population structure in the sample,
- 2-There is continuity between the ancient and modern populations leading to isolation-by-distance that is maintained across time
- 3-Isolation by distance is not uniform, suggesting that migrations were also not uniform.

Biology department

Institute of Anthropology
Palaeogenetics Group

Prof. Dr. Joachim Burger

Johannes Gutenberg-Universität Mainz
55099 Mainz
Germany

Tel. +49 6131 39-24489

j.burger@uni-mainz.dewww.uni-mainz.de

4-There was enough migration over time so that population trees with isolated branches do not explain the data.

The best-fit model is a multiregional model with independent appearance of Scythian populations followed by migration. (However, I have a some concerns about this result.)

5-Earlier populations from the Andronovo culture appear more closely related to the Eastern Scythian branch.

1-The most important claim of the paper, I believe, is the multiregional origin. The authors make a strong claim:

"ABC analyses that considered different models for the origins of Scythian populations during the Iron Age provided strong support for a multiregional origin, with eastern and western groups arising independently within their own region. There is no support for a scenario of a single origin of Scythians in either the East or the West followed by a migration wave in one direction or the other"

However, looking at Table S14, it seems like the posterior probabilities for the Western origin model are comparable to, and for some parameter choices even higher than, those of the multiregional model. In the most favorable parameter choice, the posterior for the multiregional model is only 0.76 compared to 0.24 for the Western origin model.

As I understand things, the correct interpretation is that the multiregional model is only slightly preferred, and that there is insufficient power to distinguish between the models.

Response to Reviewer1, point 1:

Although a model of western origin is the preferred model under a simple rejection method (based on a proximity criterion), the two other methods employed here (logistic regression and neural networks) are both better and more powerful for model inference in ABC. The reasons for this are multiple, but, in summary, a rejection approach is less powerful when multiple summary statistics are considered, and regression methods extract more information from the available data (see e.g. Beaumont et al. 2002, Fagundes et al. 2007, Blum & Francois 2010). Beaumont et al. (2002) wrote a seminal paper on the advantages of using regression methods over rejection methods based on proximity criterions in ABC, and we have added text to the Supplementary Information section to clarify this issue.

Concerning the fact that a western origin model can also fit the observed data:

Indeed, the model selection posteriors (0.76 for multiregional versus 0.24 for a western origin) suggest that a western model can also fit, and thus explain the observed data. But we argue that the most important criterion is actually the evaluation of the power of our model selection method to differentiate between the candidate models (see Supplementary Table 14); this analysis shows that our model selection method applied to the candidate models had a high power to distinguish between a multi-regional and a western model. The final application to empirical data thus actually does lend high support to a multi-regional model.

2-

Even though the writing is quite clear at the sentence level, the general presentation of the results made it a very difficult read for someone with my background.

Most results depend on quantitative analysis and relatively complex demographic scenarios over multiple places and time periods, but there is not a single figure in the main text illustrating the models used, the analysis strategy, or the results.

Even though the artistic depiction of a Scythian on Figure 1 provides an image of what the population may have looked like, it would have been much more useful for me to see

- an illustration of the main demographic scenarios used,
- a more detailed summary of the different cultures, populations, and sampling locations
- some of the results

Perhaps because of this lack of visual guidance, I found that the results and discussion read like a list of loosely related observations, without a clear thread.

Response to Reviewer1, point 2:

In our revised manuscript we provide more figures and visual aid for the reader to follow our approaches and conclusions. For example, in Fig.3 you will find possible demographic scenarios for the origin of the Scythian populations. Our results are displayed in Fig. 4 to 7. Additionally, we tried to be more consistent in the naming of our cultural groups and added the numbers according to the ones assigned in Fig.2 throughout the text.

3-

Consider, as a representative example, the middle paragraph on page 11. It contains a description of a number of phenotype-associated SNPs and their observed frequency in the different samples. Unfortunately, because of the small number of loci and samples, it is impossible to draw a statistically meaningful conclusion from these data. The authors conclude by stating that these results are consistent with on-going gene flow across the steppe territory. This is not untrue, but 1- this conclusion has nothing to do with the phenotypes, and 2- as far as I can tell, the data are also consistent with the absence of on-going gene flow across the steppe territory.

Unless I missed something, this paragraph neither supports nor confirms the main hypotheses. Many paragraphs similarly read like observations only loosely connected to the previous or following observations.

Response to Reviewer1, point 3:

Throughout the revision of our manuscript we tried to be more concise in the presentation of our results and conclusions drawn therefrom. We refrain from the repetition of all our results in the discussion part, which is now shorter and more focused.

4-

Another style issue is that many of the genetic analysis discussions that are provided in the main text are not self-contained.

For example, the second paragraph on page 5 reports the results of an f_4 statistic analysis. However, there is no discussion of what was the null model, and the reader has to go to the supplement to learn that the test that was done was $f_4(\text{Yamnaya, Scythian; Test, Chimp})$ for different "test" populations.

It's already demanding of the reader to understand that $f_4(\text{Yamnaya, Scythian; Test, Chimp})$ means:
1- There is a null model in which Yamnaya and Scythian form a clade relative to populations "test" and chimp.

2- The z-score represents the departure from this assumption as measured by excess similarity between "test" and one of Yamnaya and Scythian.

But the main text does not give an indication of which clades are assumed in the null model. It doesn't even mention the Yamnaya! It's fine to leave some statistical detail to the supplement, but I believe that the reader must be provided with an idea of what is being tested.

This is particularly problematic because the results of the test are not always presented in quantitatively meaningful terms: the f_4 test shows that the "East Eurasians are more closely related to the Scythian group than to the Yamnaya", however, the authors conclude that "The Scythian group has a strong genetic component that resembles current East Asian populations". I am not sure of exactly what the authors mean by "a strong genetic component", but I haven't seen evidence for what I think it means.

Response to Reviewer1, point 4:

We tried to be clearer in the presentation of our analyses and results. In the case mentioned above we rewrote the paragraph as follows:

"We used f_4 -statistics of the form $f_4(\text{Test, LBK; EHG, Mbuti})$ and $f_4(\text{Test, LBK; Han, Mbuti})$, which are zero for those Test samples that form a clade with LBK and positive for populations that have EHG- or Han-related ancestry respectively. We plotted the results against each other, which resulted in a V-shaped pattern with Yamnaya at the apex (Fig. 6). Populations with LBK-related ancestry (e.g., present-day Europeans and Sintashta) are shifted closer to zero relative to Yamnaya, indicating that they have Yamnaya ancestry. The Iron Age groups are arrayed along a cline from Yamnaya to Ami (a population of East Asian ancestry that experienced no admixture), consistent with having ancestry from these two groups."

4- Another example of the same problem, on page 7:

" We tested for continuity between the earlier (Zevakino-Chilikta and Aldy Bel; $n=26$) and the later (Pazyryk; $n=71$) Scythian period in the East using approximate Bayesian computation (ABC). These analyses revealed that they were most likely derived from one single population that was expanding over the time period considered here."

There is not enough information in the main text to understand what was done, and no reference for the reader as to where the relevant information might be.

I find it surprising to find such strong support for a single origin model without any support for the split model, given that the single-population model should be nested within a population split model. In other words, there should not be sufficient resolution to distinguish a relatively recent split between the populations. The fact that the manuscript conclusively rules out a split suggests that either the correct split model was not included in the tested models, or that the prior probabilities unfairly penalized the more complex model.

Response to Reviewer1, point 4 (4-):

It is indeed a valuable point that the prior probability for splitting time may have unfairly penalized the more complex model (with t ranging from 109–4,000 g BP). Hence, we repeated simulations with a much narrower splitting time prior (109–400 g BP) but find that the results remain unchanged: a model of genetic continuity still fits the data better than a split model. These results are now integrated into the manuscript (see Supplementary Information 1).

5-

It is hard for a non-specialist to jump back and forth between names of populations/cultures, time periods, sampling locations, sample ID, and modes of subsistence. This was a strong barrier for me throughout the manuscript.

For example, in the Admixture discussion (Figure S12), the text refers to the "ancient steppe genomes", the Scythian individuals, the European Hunter-Gatherers, and so forth, but Fig S12 uses different labels. I figured it all out eventually, but that required energy that could have been better spent elsewhere.

Response to Reviewer1, point 5:

Figure S12 is now Fig.7 and has additional labels summarizing steppe populations and hunter-gatherer. We think our description is now easier to follow, since the reader doesn't have to change between main text and SI to see the figure to the text.

6-

The authors face a very challenging sampling problem since they need to be representative in both space and time. Many estimates are based on very few genomes which may or may not be representative of broader populations. these estimates are often provided without a mention of statistical uncertainty.

For example, the first paragraph of page 10 looks at a time course of ancestry proportions over time; it would have been interesting to discuss whether these changes are statistically significant.

Similarly, the proportion of "eastern" mitochondrial lineages identified in the Aldy Bel sample is listed in the main text as "66.7%" (p.6) , and this is compared, e.g., to the proportion in modern Altaians of "65.5%". Looking through the supplement, the proportion is based on 15, so 10 out of 15 samples had an "eastern" haplotype. Reporting three significant figures for such a small sample size is misleading.

Response to Reviewer1, point 6:

During our revision the paragraph on former page 10 has been removed as well as the proportion of eastern lineages. Where proportions are still given in the text, we tried to be reasonable with the number of positions after the decimal point.

7. The analysis of the mtDNA data makes use of and MDS figure and FST, and FCT, but not of an inferred tree, which seems to be the natural way to display mtDNA sequence relatedness. Why is that?

Response to Reviewer1, point 7:

The MDS plot is a perfect aid to visualize the Reynolds' distances and the point we try to make. We don't see how a tree would be more helpful.

8-The admixture analysis using Shotgun sequencing is unconvincing (Figs S27-S29.) For example, Figure S29 shows Europeans having about 60% of the East Asian component, against 5% in S28 and 15% in S27. Given these large differences for the same individuals in different runs, why should we even bother to compare values inferred for different ancient individuals across runs?

Response to Reviewer1, point 8:

We agree and excluded the weakest sample from the analyses and combined shotgun and capture results in single figures.

9-Some of the results are split between the results and the discussion section with no obvious rationale.

For example, ABC simulations supporting continuity are presented in the results section, whereas the results supporting the mutiregional origin are presented in the discussion. This is particularly confusing because the multiregional model, presented on page 11, is necessary to understand the Andronovo discussion, presented on page 8.

Response to Reviewer1, point 9:

We agree and addressed this issue during the revision of our manuscript. We restructured the results section and rewrote the discussion part. We think that the text is now much clearer and easier to follow.

Reviewer #2 [Remarks to the Authors]:

This work describes genome wide data from 8 and mtDNA sequence from 96 Eurasian Steppe individuals and ties a demographic model of origins to a specific cultural complex. My overall opinion is that within these data and their analysis there is a basis for publication in Nature Communications. However, I have criticisms of the work as it is presented which need to be addressed.

I found it difficult to follow the results which support the inferences made in this paper, chiefly the different ancestries of east and west Scythian nomads. The text argument really must be better supported, paragraph by paragraph, by figures and tables integrated into the main text - as is allowed by this journal. Currently there are only two figures, neither of which convey genetic data or analysis! Some recommendations below:

ADMIXTURE. This is the first analysis mentioned in the discussion and one of the main ways of conveying structure in autosomal data - e.g. it has been used effectively in Allentoft et al. Here, it should be displayed, focusing on the most defensible value of k (9 perhaps). A wider analysis should be considered, are there other Iron Age genome wide samples that could be incorporated from prior data - I recall Allentoft et al, Gamba et al, perhaps Mathieson et al may have such. The mtDNA results incorporate data from the literature, but the autosomal analyses do not. Can the shotgun and capture data used here be combined into a single digram, perhaps using an ANGSD based analysis?

Response to Reviewer2, point 1:

During our revision, we combined capture and shotgun data and re-analysed them together. In the main text we now show part of the new ADMIXTURE plot with all Scythian individuals and the whole ADMIXTURE results are given as Supplementary Data Fig.1.

Shotgun coverage levels are a key part of the work and should be quoted in the main text, not referred to in a remote supplementary table.

Response to Reviewer2, point 2:

The coverage of the shotgun samples is indeed very important. It is now given in Table 1 in the main text.

PCA - a common plot for all ancient genome wide data should be shown in the main text.

Response to Reviewer2, point 3:

We agree and added two new PCA plots to the main text with combined capture and shotgun data.

f4, f3 statistics. Can these be displayed graphically in the main text. Perhaps incorporated with a map - it is difficult for a reader unfamiliar with the archaeology to translate the populations into positions in space and time as s/he reads this. Perhaps a summary table of key tests and findings, e.g. as used by Allentoft et a.

Response to Reviewer2, point 4:

We agree that the descriptions are hard to follow especially if there is no visual aid. In the revised version of our manuscript we added a figure depicting the results of our f4 analyses.

A figurative display of the mtDNA modelling results needs to be incorporated into and defended in the main text. These are key to the papers argument. Perhaps a version of the map in supp fig 11 and a cartoon of the key models used.

Response to Reviewer2, point 5:

We added visual aid in form of Fig.3, where our candidate demographic scenarios for the origin of the Scythian populations are depicted.

We found the reviewers' comments very helpful during the revision of our manuscript and we hope that we met their criticism in an appropriate fashion.

Yours sincerely,

Martina Unterländer, Friso Palstra and Joachim Burger

Beaumont, M. A., Zhang, W. & Balding, D. J. Approximate Bayesian computation in population genetics. *Genetics* 162, 2025-2035 (2002).

Fagundes, N. J. et al. Statistical evaluation of alternative models of human evolution. *Proc Natl Acad Sci U S A* 104, 17614-17619 (2007).

Blum, M. G. B. & Francois, O. Non-linear regression models for Approximate Bayesian Computation. *Stat Comput* 20, 63-73 (2010).

Dear Editor, dear referees,

We thank you for your positive and extremely useful comments. They prompted us to perform further analyses concerning the model choice procedure. These analyses helped us to make the inference on the origin of the Scythians clearer and to improve the manuscript.

You will find our point-by-point responses below; our responses to the referees' comments are indented and blue and any subsequent modifications are double indented and "quoted" in *italics*.

For the sake of clarity, we changed wording and grammar in some places that appear highlighted (yellow) in our revised manuscript (main text and SI). Additionally we shortened the *Material & Methods* section of the main text.

Reviewer #1:

The authors have made substantial improvements to the manuscript, especially regarding legibility. I still have some concerns, but I trust that they can be addressed fairly easily.

1-The authors have not satisfactorily addressed my concern about the degree of confidence in the multiregional model. The western origin model, whose posterior probability is 24%, cannot be considered "rejected" as the authors do on line 293. The argument made in the rebuttal, that the ABC has high power in simulations, is entirely unconvincing.

If the data provides similar support to two models, one cannot just consider the one with most support as strongly supported just because the same method applied to a different dataset has provided a high confidence call.

When one simulates according to the model used in inference, one naturally has higher power to identify the correct model. In fact, I suspect that the posteriors for the correct model in the simulations were stronger than the support for the multiregional model in the data.

Finally, even in the idealized context where simulations and inference model use the same model, the evidence appears weak. According to table S14, the Western Origin model would get rejected 11% of the times, and 9% in favor of the multiregion model, even if it were true. This is not even marginal significance for

rejecting a model.

The data shows only weak support for the multiregional model. The easy way to address this concern is to water down the claims accordingly throughout the manuscript. If the authors want to argue that there is strong support, then they need to provide a more compelling argument.

The referee is absolutely right in pointing out that the western origin scenario cannot be ruled out as it has ~25% posterior probability, which is not negligible at all. Supplementary Table 14 shows that the power to detect the true scenario is good for both the western and the multiregional origin model and it is also true that the information in Supplementary Table 14 cannot be used to discard the western origin scenario. But we think that referee 1 has misread Supplementary Table 14 when he states that the western origin model would get rejected 9% of the times in favour of the multiregional model, because it is only 3% in favour of the multiregional model.

Although we still think that the support for the multiregional scenario is substantially stronger (>70% posterior probability), we agree with the referee that it is important to mention that the western origin model is a reasonable alternative and we changed the text accordingly as suggested (see below).

In addition to the above clarifications, we undertook further, more detailed analyses of the relative support for the western and multi-regional models of Scythian origins. The results of these new analyses provide further support for the multiregional origin of the Scythians based on the following key arguments:

- A direct two-way comparison of the western and multi-regional models consistently shows that the multiregional model has always the stronger support across different thresholds and model selection methods (main text lines 139–145, Supplementary Information 1, Supplementary Figure 5b).
- Bayes factors associated with these two-way analyses show that the relative support for the multiregional model ranges from ‘substantial’ to ‘strong’ depending on the different thresholds and model selection methods used. (Supplementary Information 1, Supplementary Figure 5b).
- A novel model selection method based on random forests (Pudlo et al. 2016), that circumvents many of the fundamental criticisms associated with model selection under ABC (see e.g. Robert et al. 2011) provides additional support for the multiregional model by giving a posterior probability (0.672) that is very similar to that found in our initial analyses (~0.7). (Supplementary Information 1).

In accordance with the reviewer’s comment, we modified the main text and the SI to mention the western origin as an alternative model and additionally present the results of our new analyses:

lines 136–145: “According to our model selection algorithm, a multiregional model provided the best fit to the empirically-observed diversity patterns (Supplementary Table 12, 0.5% closest simulations, posterior probability

p=0.701 for logistic regression, p=0.715 for neural networks method), while a model of western origin also received some support (Supplementary Table 12, 0.5% closest simulations, posterior probability p=0.286 for logistic regression, p=0.267 for neural networks method). Therefore, contrary to the eastern origin model, a western origin cannot be fully discounted by our analysis. However, a pairwise comparison through the computation of Bayes factors reveals a strong (logistic regression) or substantial (neural network) support for the multiregional origin over the western origin model (see Supplementary Information 1, Supplementary Table 14, Supplementary Figure 5b).

lines 305–309: “However, a common origin for all of these groups either in the West or the East is unlikely (western origin) or can be rejected (eastern origin), since ABC analyses that evaluated different models for the origins of Scythian populations provided the strongest support for a multiregional origin, with eastern and western groups arising independently within their own regions.”

Supplemental Information 1:

“These inferences become more pronounced when repeating the analyses with these two candidate models only (multiregional model; 0.5% closest simulations, posterior probability p=0.917 for logistic regression, p=0.929 for neural networks method), and furthermore show that model choice is independent of tolerance rate (Supplementary Figure 5b). When computing Bayes factors between those two scenarios, support for the multiregional model ranges from ‘substantial’ to ‘strong’ for the logistic regression method, and from ‘weak’ to ‘substantial’ for the neural networks method, depending on the tolerance rate (Supplementary Figure 5b). An evaluation of confidence in the model selection procedure revealed low overall error I (3.3 %) and error II (1%) rates (Supplementary Table 14), confirming that the model selection procedure had a high power to discern scenarios and uncover the most likely one. Focusing on the two best fitting models, a western origin model had 3% chance of being mistaken for a multiregional model; vice versa a multiregional model had a 9% chance of being incorrectly identified as a western origin model. Still, model selection under ABC has been the topic of considerable debate (e.g. ³⁰) and therefore we repeated analyses using a novel approach based on random forests³¹, which confirmed that the multiregional model has the highest support (model posterior 0.672 in a two-way analysis based on a sample of $5 \cdot 10^4$ and a regression forest of 500 trees, with the R-package abcrf). Together, these analyses show that while a western origin model can also explain the observed genetic diversity patterns and cannot be completely discounted, a multiregional origin model has a higher statistical support and may thus be considered the preferred one.”

Consistently, we “watered down” the statement in the abstract and replaced the word

“reveals” by “points to” in the following sentence:

“Demographic modelling suggests independent origins for eastern and western groups with ongoing gene-flow between them, plausibly explaining the striking uniformity of their material culture”

2-lines 120-130

Similarly, the argument for continuity should be clarified. The additional simulation performed does add support to the proposed interpretation. However, the main text should specify that the split model only considered splits older than 108 generations ago. The evidence is that the two populations are unlikely to have diverged more than 108 generations (or 2.7ky) ago. This is useful evidence, but more specific than what is claimed in the main text.

This section should also point the reader to the supplement for the detailed information.

We thank the reviewer for her/his insightful suggestion. We now explicitly state that the analysis only tested a split older than 108 generations (approximately 2.7 ky BP) and make a reference to the corresponding Supplementary Information 1.

The new text reads (lines 125–132):

“These analyses revealed that they were most likely derived from one single population that was expanding over the time period considered here, i.e. the two samples are unlikely to represent two independent populations that diverged earlier than 108 generations before present (g BP) or ~2.7 ky BP. This scenario was highly supported in our model selection procedure (Supplementary Table 9, logistic regression, $p=0.995$, neural networks $p=0.568$, cf. confidence in model choice in Supplementary Table 10), whereas two scenarios that assumed that the eastern Scythian sample groups were derived from two previously diverged populations received very little statistical support (cumulative posterior probabilities: logistic regression, $p\leq 0.001$, neural networks, $p=0.001$).”

3- 1206: why does being shifted away from Yamnaya indicate Yamnaya ancestry?

We are glad the reviewers noticed this passage, which is clearly meaningless and has been deleted.

Minor points and typos, etc.

Throughout: Principal components, not principle components

We thank the reviewer and corrected this.

Reviewer #2:

The paper has been substantially altered, with more argument centred on the data and analysis in the main text.

I have some remaining criticisms.

line 129 p value of 0.000 doesn't make sense, surely $p < 0.001$

We thank the reviewer for this suggestion, and have now corrected it as suggested (line 132).

line 131-141 This paragraph is central as it describes the conclusion of a multi regional model for origins. The posterior probability of the favoured model is given as ~ 0.7 . however, the reader is not furnished with the evidence selecting this model rather than the other options. Is it significantly more likely? If so, what is the weighting in favour of it?

The reviewer is absolutely right in pointing this out. We have now expanded this paragraph to provide additional explanation, and we refer to the appropriate places in the SI where the additional supporting evidence can be found. We now additionally discuss an alternative western origin model which receives some support but substantially less in comparison to the multi-regional model.

The wording is now as follows (line 136–145):

“According to our model selection algorithm, a multiregional model provided the best fit to the empirically-observed diversity patterns (Supplementary Table 12, 0.5% closest simulations, posterior probability $p=0.701$ for logistic regression, $p=0.715$ for neural networks method), while a model of western origin also received some support (Supplementary Table 12, 0.5% closest simulations, posterior probability $p=0.286$ for logistic regression, $p=0.267$ for neural networks method). Therefore, contrary to the eastern origin, a western origin cannot be fully discounted by our analysis. However, a pairwise comparison through the computation of Bayes factors reveals a strong (logistic regression) or substantial (neural network) support for the multiregional origin over the western origin model (see Supplementary Information 1, Supplementary Table 14, Supplementary Figure 5b).”

line 202-208 The interpretation of the f_4 test is not made very clear. What is the line in figure 6 meant to signify? Why does a shift to zero (which above is taken to indicate LBK ancestry) indicate Yamnaya ancestry - line 206. Perhaps labelling key groups of populations on the plot itself might help the reader to access this information better, it is difficult from the key alone.

We agree with the reviewer and re-phrased this paragraph. It now reads (line 213–216):

“We used f_4 -statistics of the form $f_4(\text{Test}, \text{LBK}; \text{EHG}, \text{Mbuti})$ and $f_4(\text{Test}, \text{LBK}; \text{Han}, \text{Mbuti})$, which are zero for those Test samples that form a clade with LBK and positive for populations that have EHG- or Han-related ancestry respectively. We plotted the results against each other, which resulted in a V-shaped pattern with Yamnaya at the apex (Fig. 6)”

We additionally changed Fig. 6 and the legend accordingly. The legend now reads:

*“Fig. 6 **Visualisation of f-statistics results.** $f_4(\text{Test}, \text{LBK}; \text{Han}, \text{Mbuti})$ values are plotted on X axis and $f_4(\text{Test}, \text{LBK}; \text{EHG}, \text{Mbuti})$ values on Y axis, positive deviations from zero show deviations from a clade between Test and LBK. A red dashed line is drawn between Yamnaya from Samara and Ami. Iron Age populations that can be modelled as mixtures of Yamnaya and East Eurasians (like the Ami) are arrayed around this line and appear to be distinct from the main North/South European cline (blue) on the left of the X axis.”*

line 209 $f_3(\text{Test}; \text{Yamnaya_Samara}, \text{Han})$ to check whether a Test population has intermediate allele frequencies between Yamnaya_Samara and Han, which can only occur if it is a mixture of populations related to these two sources.

Is this strictly true? What about an ancestral population. Surely these populations need not be the actual sources, but proxies related to sources.

We completely agree and rephrased this paragraph. It now reads (line 219–222):

“We also computed statistics of the form $f_3(\text{Test}; \text{Yamnaya_Samara}, \text{Han})$ to check whether a Test population has intermediate allele frequencies between Yamnaya_Samara and Han, which are used as proxies for possible source populations. Intermediate allele frequencies can only occur if the test population is a mixture of populations related to these two sources”

Dear Editor,

We thank you and the reviewer for these helpful comments. You will find our point-by-point responses below. Our responses to the reviewer's comments are indented and blue and any subsequent modifications are double indented and "quoted" in *italics*.

REVIEWERS' COMMENTS:

Reviewer #1 (Remarks to the Author):

The modifications to the manuscript are a step in the right direction, yet I still have a number of concerns. Once again, these concerns do not necessarily require new analyses, but are requests for clarifications and more measured conclusions.

1-

I am still concerned about model selection and the discussion surrounding the multiregional vs western origin model. The manuscript reinterprets the same signal through through many different model selection tools. It's fine to report all the tests to show consistency, but it should be made clear that these are not independent lines of evidence.

For example, after acknowledging that the posteriors do not offer definitive evidence in support of one model versus the other, the manuscript try to strengthen the claim (l. 142):

"However, a pairwise comparison through the computation of Bayes factors reveals a strong (logistic regression) or substantial (neural network) support for the multiregional origin over the western origin model (see Supplementary Information 1, Supplementary Table 14, Supplementary Figure 5b)."

This is not additional evidence: All these methods are testing for the same signal in slightly different ways. This sentence appears to suggests that we should pay

special attention to the last two Bayes factor tests. Given that there were over 40 tests performed on Figure S5b, there should be a correction for multiple testing. If the different tests are just shown for consistency, the authors can't cherry-pick the most significant tests.

Second, the claim that support is "strong" for logistic regression and "substantial" for neural network is not supported by figure 5b: Most tests using the Neural network provide "weak" evidence (rather than "substantial"), and most tests with the logistic regression offer "substantial" support (rather than "strong"). This is reported correctly in the supplement, but not in the main text.

Third, it may be useful to provide K values together with the interpretation "weak/substantial/strong" scale. As I understand things, "strong" support would be the bar typically used to establish statistical significance (corresponding roughly to a $p=0.05$ in a frequentist approach). The additional Bayes factor analysis therefore basically repeats what had been observed already: the multiregional model has a somewhat higher posterior.

We thank the reviewer for his/her persistence concerning the point of the Scythian origin – it helps us to approach this subject more cautiously.

Regarding the first point, he/she is correct in stating that the new analyses are based on the same signal, i.e. the same set of simulations, but we do feel that the ensemble of additional tests strengthens the inference we can draw from this signal. We think that we already phrased the results carefully and in a circumspect manner, see line 127-128: "Therefore, in contrast to the eastern origin model, a western origin cannot be fully discounted by our analysis."

Concerning the second point, we changed the wording in line 128-130 in accordance with the one we also used in Supplementary Note 1:

In addition, a pairwise comparison through the computation of Bayes factors reveals a substantial to strong (logistic regression) or weak to substantial (neural network) support for the multiregional origin over the western origin model.

With regard to the third point, the K values are included in Supplementary Fig. 5b.

2-

The supplement states that:

"These inferences become more pronounced when repeating analyses with only the western and multiregional candidate models (multiregional model; 0.5% closest simulations, posterior probability $p=0.917$ for logistic regression, $p=0.929$ for neural networks method)".

Shouldn't the posteriors of the two remaining models be in proportion to their posterior probabilities

when three models were considered?

This is an interesting point. This expectation would only be true under a model selection method where we strictly use a proximity threshold (the direct method), and only use counts to estimate model posteriors. However, the logistic and neural networks regression methods function by considering the actual distribution of data points within the retained set of simulations. Model selection results can therefore readily change if data from only two models are taken into consideration, as this has the potential to change their relative distributions. Our reply therefore is: no, not necessarily.

3-

L 306: “However, a common origin for all of these groups either in the West or the East is unlikely (western origin)...”

This is not supported by the analysis. I don't think that a 27% posterior probability makes a model “unlikely”. Given the weak signal, it is also quite plausible that peculiarities of the model choice tipped the balance in favor of the multiregional model. For example, results might have been very different if direct migration were allowed from Asia into the WS in the western origin model, or gene flow between the WS and ES. It's never possible to test all models, and the manuscript does explore a reasonable set of models, but the supported conclusion is that the multiregional model was the best fit, not that a western origin is unlikely.

We agree and deleted the part of the sentence, which stated that a western origin model is unlikely. It now reads (line 259-263) as follows:

However, ABC analyses that evaluated different models for the origins of Scythian populations provided the strongest support for a multiregional origin, with eastern and western groups arising independently within their own regions.

4-

“Importantly, gene flow between the Iron Age Scythian groups was ongoing and substantial, with asymmetrical gene flow from western to eastern groups, rather than the reverse (see Supplementary Table 17 for details)” (l. 147)

and

“Despite separate origins and the enormous geographic separation, demographic modelling infers ongoing and substantial gene flow between eastern and western groups” (L 309“)

I think that the authors mean Table S15, since I do not see that Table S17 is related to this discussion.

Looking at Table S15, I do not see statistical support for asymmetric gene flow (since the confidence intervals overlap substantially).

I did not see units provided for the migration rate, so it's hard to judge about whether migration is substantial.

Finally, where was "ongoing" migration shown? I assume that the alternative is punctuated gene flow, but where was this ruled out?

Reviewer #1 is absolutely right; the wrong table was cited in this paragraph. We changed the text accordingly; the reference is now in line 135 and to Supplementary Table 13 and 15.

Supplementary Table 13 also shows that a model of two-way continuous gene flow outperformed models where gene flow is limited to single directions and shorter time spans. However, it is true that we did not consider punctuated gene flow models in all our analyses. We thus changed the wording slightly to address this concern (line 134-137).

"Importantly, our simulations support a continuous gene flow between the Iron Age Scythian groups, with indications of asymmetrical gene flow from western to eastern groups, rather than the reverse (see Supplementary Table 13 and 15 for details)."

5-

"The two Early Sarmatian samples from the West (group #3 in Fig. 2) cluster with an Iron Age sample from the Samara district and are generally close to the Early Bronze Age Yamnaya samples from Samara and Kalmykia and the Middle Bronze Age Poltavka samples from Samara. The eastern samples from Pazyryk (#6), Aldy Bel (#5), and Zevakino-Chilikta (#4) are part of a loose cluster with other samples from Central Asia, including those from Okunevo, Late Bronze Age and Iron Age Russia, and Karasuk."

"Cluster with" is an overstatement. It also seems to me that the Karasuk are closer to the Early Sarmatian than are groups #4-6.

It would be useful to have a main conclusion to this paragraph.

We replaced "cluster with" by "fall close to" in line 158.

The main conclusion of this paragraph is that, in Figure 4, Iron Age nomads generally form a loose cluster with individuals of geographical proximity: e.g. Early Sarmatians fall close to Iron Age samples from southern Russia ("Samara_IA"), and eastern Scythians cluster with samples from Central Asia, including Karasuk and Okunevo individuals. We would prefer not

to add another concluding sentence to this paragraph, as the sentence already represents a summary.

6-

“Since the PCA of west Eurasia in Figure 4 does not allow one to examine the relationship of the ancient samples to contemporary world populations, we also carried out principal component analysis of all 2,345 modern individuals in the Human Origins dataset,”

I do not understand why “the PCA of West Eurasia in Figure 4 does not allow one to examine relationships [to present-day populations]”. After all, this was performed by projecting the ancient samples onto PCAs formed by present-day populations. Wouldn't it simply be a matter of coloring the gray dots? I don't see how adding Africans to the PCA helps resolve anything about the ancient samples. It would only hide the finer-scale relationships, wouldn't it?

We agree in principle. However, as the first principal components of a PCA are not necessarily a direct proxy for the demographic processes that shaped them and can be biased by data quantity, it makes sense to view them with different data sets and at varying resolutions. Figure 4 serves to illustrate the genetic relationship between ancient individuals. Figure 5, on the other hand, tackles a completely different point, namely the positioning of the Scythians between West and East Eurasians. It demonstrates that the Iron Age nomads are arrayed between West Eurasian and East Asian populations, which is not visible on the small scale PCA of West Eurasians in Figure 4. To further clarify this point, we changed the wording in line 174-175:

Since the PCA of west Eurasia in Figure 4 does not allow one to examine the ancient samples in relation to contemporary East Asian populations, ...

7-

L 217 “The Iron Age Scythians are arrayed along a cline from Yamnaya to Ami (a population of East Asian ancestry that experienced no admixture), consistent with having ancestry from populations genetically similar to these two groups.”

L 228 “Iron Age populations that can be modelled as mixtures of Yamnaya and East Eurasians”

Unless I am missing something, this is a very weak argument: There is no obvious cline along the red line, nor is there obvious “arraying”. I see a line that was drawn between two arbitrarily chosen points, and that happens to fall close to another set of points.

It's also unclear to me whether a mixture population would indeed fall halfway between the source populations on this graph.

Figure 6 does suggest that the Iron Age populations are somewhat more closely related to the Asian populations than the European populations, but this is a strange way to go about showing this. I don't think that it adds information beyond what is already visible from Figure 5.

Figure 6 is a visual representation of two f_4 statistics and we show formally that the populations along the red line can be modelled as East/West Eurasian admixtures using qpWave/qpAdm (line 206-220).

This test is a generalization of the f_4 ratio (Patterson et al. 2012 – Ancient admixture in human history. *Genetics* 192, 1065-1093), and tests whether a *Test* population is intermediate between putative ancestral sources, using all pairs of considered outgroups (not just Han, EHG and Mbuti used here for the purposes of visual illustration).

Figure 6 shows that Iron Age individuals can be modelled as a mixture of Yamnaya and East Asians but not Europeans and East Asians. The information content of this figure is therefore different to that of Figure 5.

8-

L297: “Thus, the results of the examination of phenotypic SNPs that show frequency differences between Europe and East Asia are consistent with on-going gene flow across the steppe territory.”

I'm not sure how this conclusion is reached. I agree that the observations are not inconsistent with on-going gene flow, but they do not really support gene flow, let alone “ongoing” gene flow. Why is the consistency of phenotypic SNPs to an ongoing gene flow model relevant?

As shown in Supplementary Table 28, several SNPs with evidence of selection in either modern European or Asian populations are present in ancient Scythian population from both, the East and the West. For example, derived alleles of the three “pigmentation markers” *HERC2*, *SLC24A5* and *SLC45A2* that are thought to have been under strong selection in Europeans, were detected in eastern Scythians. As ancient Scythians very generally display a mosaic of European and Asian-associated genotypes, we argue that gene flow is the most likely explanation – which is consistent with other lines of evidence.

However, we agree that there is no evidence for “ongoing” gene flow; accordingly we deleted the word “ongoing” in line 251.

9-

L322: “In general, gene-flow between eastern and western Eurasia seems to have been more intense during the Iron Age than in modern times”

This sounds plausible, but I do not see where this was discussed.

This argument is mainly based on AMOVA analysis, as well as on the lower F_{ST} values between ancient populations than those observed between modern populations from similar regions; we describe these results in lines 223-228. While we think that this is strong evidence, we refrained from further interpretation, as we believe that this issue deserves further investigation on the basis of a denser sampling in the future.

More minor points

10-

I. 172: “We find very high mitochondrial haplotype diversity in our sample set, ranging from 0.958 ± 0.036 in the Tagar/Tes sample (group #7 in Fig. 2) up to 1.000 ± 0.039 in the Early Sarmatians

Very high with respect to what?

The observed haplotype diversity is very high in absolute terms. Values close to 1 indicate that almost every individual carries a different haplotype. Since the lower margin of estimated values is not completely uncommon in some modern populations of Central Asia, we changed the wording in line 156-158 to:

The mitochondrial haplotype diversity in our sample set ranges from 0.958 ± 0.036 in the Tagar/Tes sample (group #7 in Fig. 2) up to 1.000 ± 0.039 in the Early Sarmatians (#3; Supplementary Table 7).

11-

“These samples contrast with earlier samples from the Eurasian steppe belonging to the Andronovo, Sintashta and Srubnaya groups, which overlap Late Neolithic/Bronze Age individuals from mainland Europe and are shifted ‘southwards’ in the PCA plot towards the early farmers of Europe and Anatolia.”

Is “southwards” used to refer to the bottom of the PCA figure? If so, using geographical terms for this purpose is confusing. If it refers to geography, please explain the mapping.

We agree that the word “southwards” is misleading, and therefore changed it to “downwards” (line 169).

12-

Figure 4 has no axes labels.

We changed Figure 4 and 5. The axes of the PCAs now have labels.

13-

Supplementary Table 11 does not appear to be quoted in the text.

We added a reference to this table and its content in line 115-116.